# Adaptive Flow Matching for Resolving Small-Scale Physics

**Stathi Fotiadis**[* 1]  **Noah D Brenowitz**[2]  **Tomas Geffner**[2]  **Yair Cohen**[2]  **Michael Pritchard**[2]  **Arash Vahdat**[† 2]  **Morteza Mardani**[† 2]

[*]Work done during an internship at NVIDIA.
[†]Equal advising.

## Abstract

Conditional diffusion and flow-based models can super-resolve small-scale details in natural images, but weather and other physical domains face three unique challenges: $(i)$ spatially misaligned inputs and outputs (different PDE resolutions yield divergent trajectories), $(ii)$ mismatched and distinct input-output channels (channel synthesis), and $(iii)$ channel-dependent stochasticity. To address these challenges, we first encode the inputs into a latent base distribution that reconstructs the large-scale and more deterministic parts of the target. Next, we handle the remaining uncertainty by injecting noise via an adaptive noise scaling mechanism, informed by maximum-likelihood estimates of the encoder's RMSE. Finally, we apply Flow Matching to refine the latent state and add fine-scale physics. Experiments on real-world weather data (e.g., 25 km to 2 km super-resolution in Taiwan) and synthetic Kolmogorov flow indicate that our Adaptive Flow Matching (AFM) framework provides improvements over prior baselines—particularly for more stochastic channels—and consistently achieves better-calibrated ensembles.

## 1. Introduction

Resolving small-scale physics is crucial in many scientific applications (Wilby et al., 1998; Rampal et al., 2022; 2024). For instance, in atmospheric sciences, accurately capturing small-scale dynamics is essential for local planning and disaster mitigation. The success of *conditional* diffusion models in super-resolving natural images and videos (Song et al., 2021; Batzolis et al., 2021; Hoogeboom et al., 2023)

has recently been extended to super-resolving small-scale physics (Aich et al., 2024; Ling et al., 2024). However, this task faces significant challenges: ($C$1) Input and target data are often *spatially* misaligned due to distinct PDE solutions operating at various resolutions, leading to divergent trajectories. For instance, the eye-of-typhoon is spatially misaligned between low and high resolution in CWA data due to different dynamical models used to simulate each scale (see Figure 1). Additionally, the input and target variables (channels) often represent different physical quantities, causing further misalignment. ($C$2) the data exhibit multiscale dynamics, where certain large-scale processes are more deterministic (e.g. the propagation of midlatitude storms), while small-scale physics, such as thunderstorms, are highly stochastic; and ($C$3) the record of Earth observations is limited compared to natural image datasets.

Few efforts have been made to directly address these challenges in generative learning. Prior work typically relies on *residual* learning approaches (Mardani et al., 2023). For instance, the method proposed in (Mardani et al., 2023) introduces a two-stage process where the deterministic component is first learned through regression, followed by applying diffusion on the residuals to capture the small-scale physics. While this approach offers a way to separate deterministic and stochastic components, it poses a significant risk of overfitting. The initial regression stage may overfit the training data, and thus fails to adequately represent the variability of the small-scale dynamics when training the diffusion model in the second stage (see Appendix A). Additionally, this two-stage method lacks a principled way to handle the uncertainty inherent in both the deterministic and stochastic components.

To address these limitations, we propose an end-to-end approach based on Flow Matching. The key elements of our method are as follows: First, an encoder maps the coarse-resolution input data to a *latent* space that is closer to the target fine-resolution distribution. Flow matching is then applied starting from this encoded distribution to generate the target distribution. The encoder captures the deterministic component, which is then augmented with noise to in-

[1]Imperial College London [2]NVIDIA. Correspondence to: Morteza Mardani <mmardani@nvidia.com>.

*Proceedings of the 42$^{nd}$ International Conference on Machine Learning*, Vancouver, Canada. PMLR 267, 2025. Copyright 2025 by the author(s).

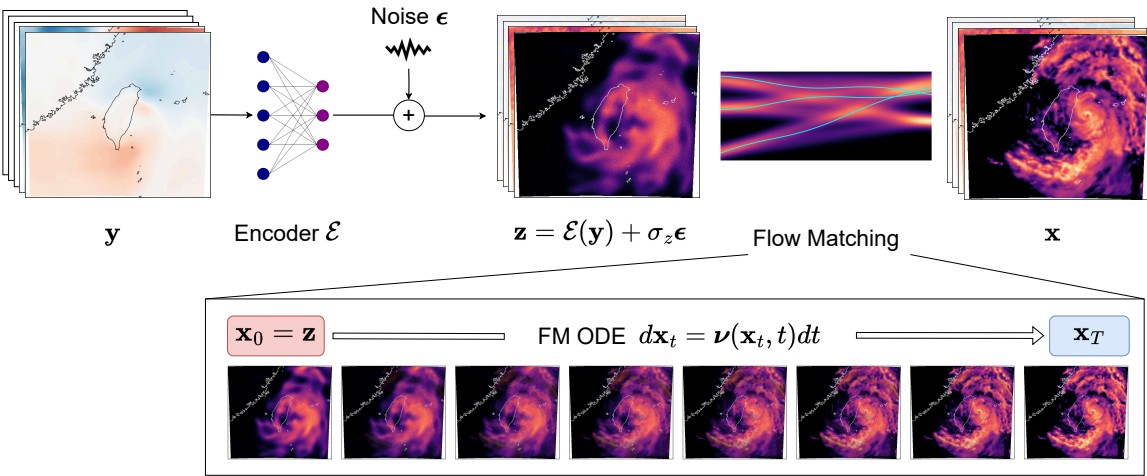

Figure 1: **Overview of Adaptive Flow Matching (AFM).** The encoder transforms (coarse-res.) inputs into a latent distribution more aligned with the (fine-res.) target. It generates channels absent in the input and corrects both *spatial* and *channel* misalignments, such as repositioning the typhoon's eye to its more accurate location, and generating radar data. From the latent space, FM generates small-scale physics by transporting samples from $p(\mathbf{z})$ to $p(\mathbf{x})$ via the velocity field $\boldsymbol{\nu}(\mathbf{x}, t)$.

troduce uncertainty. The deterministic prediction is based on the idea that physical processes occur on different time scales, with larger-scale physics having longer-term, more deterministic effects. We refer to this method as *Adaptive Flow Matching (AFM)*, where the stochasticity is controlled by the noise injected at the encoder output. Proper tuning of the noise scale is critical to balance deterministic and stochastic dynamics. To achieve this, we propose a maximum likelihood procedure that adjusts the noise scale based on the encoder's error, dynamically tuning it *on the fly* according to the encoder mismatch.

AFM can be viewed through the lens of diffusion models, efficiently implemented using a stable denoising objective. We conduct extensive experiments on both synthetic and real datasets. For the real data, we use the same data as Mardani et al. (2023), the best estimates of the 25km and 2km observed atmospheric state available from meteorological agencies, centered around a region containing Taiwan. Additionally, we synthesize dynamics from a multi-resolution variant of 2D fluid-flow, where we can control the degree of misalignment. Our results show that AFM consistently outperforms existing methods across various skill metrics. Overall , our *main contributions* are summarized as:

- **Adaptive Flow Matching (AFM)**: A method for matching *spatially* misaligned data (plus misaligned *channels*) with multiscale physics.

- **Adaptive Noise Scaling**: Derived based on ML principle, which optimally balances the learning of deterministic and stochastic components between the encoder and Flow Matching.

- **Experiments on Real and Synthetic Data**: Our results show AFM outperforms existing alternatives. As

the degree of misalignment grows, conditional diffusion/flow models become progressively worse, while AFM with adaptive noise scaling still performs well.

## 2. Related Work

**Conditional diffusion and flow models**. Conditioning is a powerful technique for paired image-to-image translation in diffusion/flow models (Batzolis et al., 2021; Kawar et al., 2022; Xingjian & Xie, 2023). It is commonly used in image restoration tasks such as super-resolution and deblurring, where the goal is to map a low-quality input to a high-quality target, with corresponding pixel associations between input and target. However, as our experiments demonstrate (see Section 5), plain conditional models often fail when the data is severely misaligned.

**Diffusion Bridges and Stochastic Interpolants**. Diffusion bridges (De Bortoli et al., 2024; Shi et al., 2023; Liu et al., 2023; Pooladian et al., 2023) facilitate transitions between distributions but rely on strong assumptions, such as local alignment in I$^2$SB (Liu et al., 2023), unsuitable for *misaligned* data. Stochastic interpolants (Albergo et al., 2023; Albergo & Vanden-Eijnden, 2023; Lipman et al., 2022; Liu et al., 2022) assume smooth transport and are often trained with independent coupling between noise and data, even for conditional problems. In particular, (Albergo et al., 2023) couples base and target distributions. Chen et al. (2024) applies stochastic interpolants for fluid-flow forecast. But, our AFM suits scenarios with misaligned input-output variables, where the encoder learns large-scale deterministic dynamics, while adaptively balancing the contribution of deterministic and stochastic components.

**Co-training generative models with encoders**. Co-

Table 1: Comparison between AFM and alternatives for learning the generative map between *misaligned* data $(\mathbf{y}, \mathbf{x})$. Define $\boldsymbol{\epsilon} \sim \mathcal{N}(0, 1)$, $\boldsymbol{e} := (\mathcal{E}(\mathbf{y}) - \mathbf{x})/\sigma_z$, and $\mathbf{r} := \mathbf{x} - \mathbb{E}[\mathbf{x}|\mathbf{y}]$. The noise scale $\sigma_z$ is the ML noise estimate ensuring $\mathbb{E}[\|\boldsymbol{e}\|^2] = 1$. CDM and CFM represent conditional diffusion and flow models, respectively.

| Scheme | Perturbation Kernel | Score | Train Loss (Denoising) |
|--------|---------------------|-------|------------------------|
| CFM | $\mathbf{x}_t = (1-t)\boldsymbol{\epsilon} + t\mathbf{x}$ | $\nabla_{\mathbf{x}_t} \log p_t(\mathbf{x}_t|\mathbf{y})$ | $\mathbb{E}\big[\|\mathcal{D}_{\boldsymbol{\theta}}(\mathbf{x}_t; t) - \mathbf{x}\|^2\big]$ |
| CDM | $\mathbf{x}_t = \mathbf{x} + \sigma_t\boldsymbol{\epsilon}$ | $\nabla_{\mathbf{x}_t} \log p_t(\mathbf{x}_t|\mathbf{y})$ | $\mathbb{E}\big[\|\mathcal{D}_{\boldsymbol{\theta}}(\mathbf{x}_t; \sigma_t) - \mathbf{x}\|^2\big]$ |
| CorrDiff | $\mathbf{r}_t = \mathbf{r} + \sigma_t\boldsymbol{\epsilon}$ | $\nabla_{\mathbf{r}_t} \log p_t(\mathbf{r}_t|\mathbf{y})$ | $\mathbb{E}\big[\|\boldsymbol{e}\|^2\big] \to \mathbb{E}\big[\|\mathcal{D}_{\boldsymbol{\theta}}(\mathbf{r}_t; \sigma_t) - \mathbf{r}\|^2\big]$ |
| **AFM (Ours)** | $\mathbf{x}_t = \mathbf{x} + \sigma_t(\boldsymbol{e} + \boldsymbol{\epsilon})$ | $\nabla_{\mathbf{x}_t} p_t(\mathbf{x}_t)$ | $\mathbb{E}\big[(\sigma_z/\sigma_t)^2\|\mathcal{D}_{\boldsymbol{\theta}}(\mathbf{x}_t; \sigma_t) - \mathbf{x}\|^2 + \lambda\|\boldsymbol{e}\|^2\big]$ |

training encoders with diffusion models has been explored in various domains. DiffCast (Yu et al., 2024) uses an encoder to predict the mean precipitation for nowcasting with a diffusion model that handles the residuals. Similarly, Grad-TTS (Popov et al., 2021) and Bridge-TTS (Chen et al., 2022) integrate encoders with diffusion-based processes for text-to-speech synthesis. These methods focus on temporal generation and operate on *single channel*. In contrast, our AFM framework tackles superresolution and channel synthesis with spatial misalignment for mulitvariate data with different stochastic charactristics.

**Atmospheric super-resolution (downscaling).** In atmospheric sciences, recovery of the fine-resolution from the coarse resolution is known as *downscaling*. Several works have explored statistical downscaling; see, e.g., (Rampal et al., 2024; Wilby et al., 1998; Rampal et al., 2022). For instance, CorrDiff (Mardani et al., 2023) learns a diffusion model on residuals left from a deterministic prediction. It a two-stage approach where the overfitting of the regression error propagates to the diffusion in second stage, leading to reduced generation quality with severely under-calibrated ensembles. In AFM this is mitigated using the balance between the deterministic and stochastic errors and the adaptive noise scaling per channel.

## 3. Background and Problem Statement

Consider the task of learning the conditional distribution $p(\mathbf{x}|\mathbf{y})$ from a finite set of paired data $\{(\mathbf{y}_i, \mathbf{x}_i)\}_{i=1}^N$, where $\mathbf{y} \in \mathbb{R}^{c \times h \times w}$ and $\mathbf{x} \in \mathbb{R}^{C \times H \times W}$ represent the input and target, respectively. For example, in atmospheric sciences, $\mathbf{y}$ is a coarse-resolution forecast from the Global Forecast System (GFS) at 25 km resolution, while $\mathbf{x}$ is the fine-resolution target from a high resolution regional weather simulation system at 2 km.

This task is particularly challenging because the pairs $(\mathbf{y}, \mathbf{x})$ are often misaligned. First, these pairs might represent solutions to partial differential equations (PDEs) at significantly different spatial and temporal discretizations. This can lead to completely different temporal or spatial trajectories, including due to effects of internal chaotic dynamics. Second, the input and target may involve several different channels, e.g., corresponding to distinct atmospheric variables with intrinsic correlations, that further

challenges learning the joint distribution.

Before delving into details, it is useful to review Flow Matching as a critical component of our approach.

**Flow Matching.** It transports samples from a source distribution $p_0(\mathbf{x})$ to a target distribution $p_1(\mathbf{x})$ by learning a velocity field. Flows are often trained using a linear interpolant between noise and data. The forward process simply follows the ODE:

$$\frac{d\mathbf{x}_t}{dt} = \boldsymbol{\nu}(\mathbf{x}_t, t), \tag{1}$$

where $\boldsymbol{\nu}(\mathbf{x}_t, t)$ represents the velocity field over time $t \in [0, 1]$. For the linear interpolant, the true velocity that generates a single data sample $\mathbf{x}_1$ is given by $\boldsymbol{\nu}_{\text{true}}(\mathbf{x}_t, t) = \mathbf{x}_1 - \mathbf{x}_0$. The goal is to minimize the discrepancy between the learned velocity field $\boldsymbol{\nu}_{\theta}(\mathbf{x}_t, t)$ and the true velocity as:

$$\min_{\theta} \mathbb{E}_{t, \mathbf{x}_t} \left[ \left\| \boldsymbol{\nu}_{\theta}(\mathbf{x}_t, t) - (\mathbf{x}_1 - \mathbf{x}_0) \right\|^2 \right], \tag{2}$$

where $\mathbf{x}_t := (1-t)\mathbf{x}_0 + t\mathbf{x}_1$. Upon convergence, the learned velocity is used to generate samples $\mathbf{x}_0 \sim p_0(\mathbf{x})$ by solving the ODE in Eq. 1.

It is also useful to recognize the connection between Flow Matching and denoising diffusion models. This is useful as one can utilize the effective and scalable (denoising) training of diffusion models such as the celebrated elucidated diffusion models (EDM) (Karras et al., 2022) to stably learn velocity fields. we deffer the discussion and connections to the Appendix (see section C).

## 4. Adaptive Flow Matching

To learn the conditional distribution $p(\mathbf{x}|\mathbf{y})$, a natural approach is to use conditional diffusion/flow models. They have succeeded in image-to-image tasks e.g., image restoration or super-resolution, where conditioning provides rich information about the target (Saharia et al., 2022). However, traditional methods struggle when the input $\mathbf{y}$ and target $\mathbf{x}$ are significantly misaligned (see Section 5 for evidence). To address this, we propose a multiscale approach:

**Deterministic Dynamics:** The input $\mathbf{y}$ is encoded into a latent variable $\mathbf{z} = \mathcal{E}(\mathbf{y})$. This encoding serves two purposes: $(i)$ it first matches the large-scale, mainly determin-

**Algorithm 1** AFM training

1: **Input:** $\lambda$, $\{(\mathbf{y}_i, \mathbf{x}_i)\}_{i=1}^N$, $M$
2: Initialize $\sigma_z$, $\boldsymbol{\theta}$, $\mathcal{E}$
3: **repeat**
4:     Sample $\sigma_t \sim \mathcal{U}[0, \sigma_z]$ and $\boldsymbol{\epsilon} \sim \mathcal{N}(0, \mathbf{I})$
5:     Compute error: $\boldsymbol{e} := (\mathcal{E}(\mathbf{y}) - \mathbf{x})/\sigma_z$
6:     Perturb input: $\mathbf{x}_t = \mathbf{x} + \sigma_t(\boldsymbol{e} + \boldsymbol{\epsilon})$
7:     Take a gradient step on:
8:     $\nabla_{\boldsymbol{\theta}, \mathcal{E}} \left[ \left( \frac{\sigma_z}{\sigma_t} \right)^2 \|\mathcal{D}_{\boldsymbol{\theta}}(\mathbf{x}_t, \sigma_t) - \mathbf{x}\|^2 + \lambda \|\boldsymbol{e}\|^2 \right]$
9:     Every $M$ steps: $\sigma_z^{\text{cur}} = \sqrt{\mathbb{E}[\|\mathbf{x}_{val} - \mathcal{E}(\mathbf{y}_{val})\|^2]}$
10:                  $\sigma_z \leftarrow (1 - \beta)\sigma_z + \beta \sigma_z^{\text{cur}}$
11: **until** convergence

**Algorithm 2** AFM sampling

1: **Input:** $\mathbf{y}$, $\Delta t$, $\sigma_z$, $\mathcal{D}_{\boldsymbol{\theta}}$, $\mathcal{E}$
2: Sample noise $\boldsymbol{\epsilon} \sim \mathcal{N}(0, \mathbf{I})$
3: Form latent $\mathbf{z} = \mathcal{E}(\mathbf{y}) + \sigma_z \boldsymbol{\epsilon}$
4: Initialize $\mathbf{x}_0 = \mathbf{z}$
5: **for** $t = 0 : \Delta t : 1$ **do**
6:     $\sigma_t = (1 - t)\sigma_z$
7:     $\boldsymbol{\nu}_{\theta}(\mathbf{x}_t, t) = (\mathcal{D}_{\boldsymbol{\theta}}(\mathbf{x}_t, \sigma_t) - \mathbf{x}_t)/(1 - t)$
8:     $\mathbf{x}_{t+\Delta t} = \mathbf{x}_t + \boldsymbol{\nu}_{\theta}(\mathbf{x}_t, t) \cdot \Delta t$
9: **end for**
10: **return** $\mathbf{x}_1$

$(1 - t)\sigma_z$, the residual error by $\boldsymbol{e} := (\mathcal{E}(\mathbf{y}) - \mathbf{x})/\sigma_z$, and the noise $\boldsymbol{\epsilon} \sim \mathcal{N}(0, 1)$, the Flow Matching for joint training of the encoder and flow reduces to the denoising objective:

$$\min_{\mathcal{E}, \boldsymbol{\theta}} \mathbb{E}_{\mathbf{x}, \mathbf{y}, \sigma_t \sim \mathcal{U}[0, \sigma_z]} \left[ (\sigma_z/\sigma_t)^2 \|\mathcal{D}_{\boldsymbol{\theta}}(\mathbf{x}_t, \sigma_t) - \mathbf{x}\|^2 \right] \quad (6)$$

Intuitively, this denoising objective addresses not only the Gaussian noise typical of diffusion models but also residual errors introduced during the deterministic encoding process. Note that the residual error $\boldsymbol{e}$ conveys the essential information about the input conditioning $\mathbf{y}$ required for generating the target $\mathbf{x}$. Therefore, it is crucial to carefully balance the influence of this error by appropriately selecting $\sigma_z$ and applying regularization to the encoding process. These considerations will be discussed next.

### 4.2. Adaptive Noise Scaling

It is essential to tune the noise parameter $\sigma_z$ based on the data before applying diffusion denoising. Specifically, we consider the latent variable:

$$\mathbf{z} = \mathcal{E}(\mathbf{y}) + \sigma_z \boldsymbol{\epsilon}, \quad (7)$$

where we observe $\mathbf{x}$ and $\mathbf{y}$, and aim to adjust $\sigma_z$ in a maximum likelihood (ML) sense so that $\mathbf{z}$ aligns closely with $\mathbf{x}$. In this context, $\sigma_z$ controls the scale of the noise added to the encoder's output $\mathcal{E}(\mathbf{y})$. By leveraging the ML estimator for $\sigma_z$, we can derive it as the root-mean-square-error (RMSE) of the unnormalized residual error, namely

$$\sigma_z = \sqrt{\mathbb{E}[\|\mathbf{x} - \mathcal{E}(\mathbf{y})\|^2]}. \quad (8)$$

In practice, to prevent overfitting, $\sigma_z$ is progressively updated using the encoder's RMSE on a validation set. Intuitively, if the deterministic regression model overfits to a the training dataset, the validation RMSE will grow, and thus our AFM scales up the noise in the base distribution for the Flow Matching. During training we update $\sigma_z$ using an exponential moving average (EMA) every $M = 10k$ training steps to avoid sudden changes in stochasticity:

$$\sigma_z \leftarrow (1 - \beta)\sigma_z + \beta \sigma_z^{\text{cur}}. \quad (9)$$

istic dynamics of the input and output, aligning the spatially misaligned large-structures due to diverging trajectories, and $(ii)$ it aligns the *channels* by projecting the input into the same space as the output (since $\mathbf{y}$ and $\mathbf{x}$ represent different weather variables).

**Generative Dynamics:** Flow matching is then used to transform the base distribution $p(\mathbf{z})$ into the target distribution $p(\mathbf{x})$. To account for uncertainty in the encoding phase, we perturb the latent variable with Gaussian noise:

$$\mathbf{z} = \mathcal{E}(\mathbf{y}) + \sigma_z \boldsymbol{\epsilon}, \quad \boldsymbol{\epsilon} \sim \mathcal{N}(0, \mathbf{I}). \quad (3)$$

In the following sections, we detail the learning process for both the encoder and Flow Matching.

### 4.1. Training

The objective is to *jointly* learn the encoder and the Flow Matching model. To achieve this, we delve into Flow Matching in the latent space. Specifically, it establishes a linear interpolant defined as $\mathbf{x}_t = (1 - t)\mathbf{z} + t\mathbf{x}$, where $\mathbf{z}$ is the encoded state and $\mathbf{x}$ is the target, for $t \in [0, 1]$. Consequently, based on Equation (2), the Flow Matching objective is formulated as:

$$\min_{\mathcal{E}, \boldsymbol{\theta}} \mathbb{E}_{t, \mathbf{x}, \mathbf{z} \sim \mathcal{N}(\mathcal{E}(\mathbf{y}), \sigma_z)} \left[ \|\boldsymbol{\nu}_{\theta}(\mathbf{x}_t, t) - (\mathbf{x} - \mathbf{z})\|^2 \right]. \quad (4)$$

Due to the stochasticity in $\mathbf{z}$ and the stability of the EDM framework for training diffusion models—along with its tuning-free characteristics—it is advantageous to incorporate denoising through EDM when training Flow Matching. To this end, the first step is to recast the linear interpolant as a Gaussian diffusion process (cf. Equation (3)):

$$\mathbf{x}_t = (1 - t)\mathcal{E}(\mathbf{y}) + t\mathbf{x} + (1 - t)\sigma_z \boldsymbol{\epsilon}, \quad t \in [0, 1] \quad (5)$$

The following proposition simplifies the task of learning the velocity field as a denoising process.

**Proposition 1.** *For the perturbation model* $\mathbf{x}_t = \mathbf{x} + \sigma_t \boldsymbol{e} + \sigma_t \boldsymbol{\epsilon}$, *where the noise standard deviation is given by* $\sigma_t :=$

An illustration of the evolving values of $\sigma_z$ during training is provided in Figure 5. Once training completes we fix the final value of $\sigma_z$ for inference.

**Alternative Stochastic Encoders.** One can alternatively model uncertainty using a VAE. It essentially imposes KL regularization on the encoder output to predict $\mu$ and $\sigma$, where $\sigma$ can drive the noise for FM. While valid, we opted for a simpler ML method that offers a closed-form and intuitive solution for $\sigma$ (see (Rybkin et al., 2021)). Although the VAE allows learning of $\sigma$ from the data, it requires tuning the KL regularization and suffers from the prior hole problem.

**Deterministic to Stochastic Regime.** If the data is purely deterministic, encoder perfectly matches the data and thus $\sigma_z = \sigma_t = 0$. This means there is no work left for the generative FM . Alternatively, if data is purely stochastic, encoder returns large error ($\sigma_z, \sigma_t \gg$), which leads to base noise distribution with large variance, that triggers FM.

### 4.3. Encoder Regularization

Another effective approach to control the residual error $e$ is through regularization. Specifically, one can impose a reconstruction regularization on the encoder, encouraging the encoder output to approximate the target, i.e., $\mathbf{x} \approx \mathcal{E}(\mathbf{y})$, thus minimizing the residual error $e$. In an ideal scenario, this would lead to $e \approx 0$. However, enforcing perfect matching can adversely affect the generalization capability of the model, leading to overfitting. To mitigate this, a *soft* regularization is applied, controlled by a penalty weight $\lambda$. This weight balances the trade-off between reducing the residual error and maintaining the generalization ability of the model. The resulting objective function becomes:

$$\min_{\mathcal{E}, \boldsymbol{\theta}} \mathbb{E}_{\mathbf{x}, \mathbf{y}, \sigma_t \sim \mathcal{U}[0, \sigma_z]} \left[ \left( \frac{\sigma_z}{\sigma_t} \right)^2 \left\| \mathcal{D}_{\boldsymbol{\theta}}(\mathbf{x}_t, \sigma_t) - \mathbf{x} \right\|^2 + \lambda \left\| e \right\|^2 \right]$$

### 4.4. Sampling

Once the velocity field $\boldsymbol{\nu}_\theta(\mathbf{x}_t, t) = (\mathcal{D}_{\boldsymbol{\theta}}(\mathbf{x}_t, \sigma_t) - \mathbf{x}_t)/(1 - t)$ is learned using Algorithm 1, sampling requires integrating the flow forward in time based on the ODE in Equation (1). The forward integration process from $t = 0$ to $t = 1$ is expressed as:

$$\mathbf{x}_1 = \int_{t=0}^{t=1} \boldsymbol{\nu}_\theta(\mathbf{x}_t, t) \, dt \tag{10}$$

In practice, this integration can be approximated using Euler steps, which are detailed in Algorithm 2

## 5. Experiments

We evaluate performance of the proposed Adaptive Flow Matching (AFM) model on two datasets: i) A regional weather downscaling dataset with real-world meteorological observations from Taiwan's Central Weather Administration (CWA) ii) a synthetic multiscale Kolmogorov flow

Table 2: **AFM vs. baselines for CWA downscaling:** Values in **bold** show the best performance. AFM outperforms baselines except for the *deterministic* temperature variable, where CorrDiff excels. Temperature, being the most deterministic, benefits from CorrDiff's fully deterministic predictions. While AFM could match CorrDiff using a UNet encoder and higher $\lambda$, this would compromise stochastic predictions for other variables.

| Variable | Metric | CorrDiff | CFM | CDM | UNet | AFM |
|---|---|---|---|---|---|---|
| **Radar** | RMSE $\downarrow$ | 5.08 | 5.06 | 5.70 | 5.09 | **4.90** |
| | CRPS $\downarrow$ | 1.89 | 1.88 | 2.39 | - | **1.78** |
| | MAE $\downarrow$ | 2.50 | 2.46 | 2.78 | 2.50 | **2.42** |
| | SSR $\to$ 1 | 0.38 | 0.33 | **0.46** | - | 0.44 |
| **Temp.** | RMSE $\downarrow$ | **0.83** | 0.93 | 0.95 | 0.86 | 1.00 |
| | CRPS $\downarrow$ | **0.50** | 0.58 | 0.54 | - | 0.52 |
| | MAE $\downarrow$ | **0.60** | 0.72 | 0.70 | 0.62 | 0.67 |
| | SSR $\to$ 1 | 0.36 | 0.41 | **0.52** | - | 0.47 |
| **E. Wind** | RMSE $\downarrow$ | 1.47 | 1.45 | 1.62 | 1.49 | **1.44** |
| | CRPS $\downarrow$ | 0.85 | 0.82 | 0.93 | - | **0.80** |
| | MAE $\downarrow$ | 1.07 | **1.06** | 1.24 | 1.09 | 1.07 |
| | SSR $\to$ 1 | 0.43 | 0.50 | **0.61** | - | **0.61** |
| **N. Wind** | RMSE $\downarrow$ | 1.66 | **1.61** | 1.84 | 1.66 | **1.61** |
| | CRPS $\downarrow$ | 0.95 | 0.90 | 1.06 | - | **0.88** |
| | MAE $\downarrow$ | 1.20 | **1.16** | 1.41 | 1.21 | 1.17 |
| | SSR $\to$ 1 | 0.41 | 0.49 | **0.58** | - | **0.58** |

dataset, designed to capture variable degrees of misalignment. Both datasets present significant downscaling challenges with local and large-scale misalignment, mixed-scale dynamics, and channel-specific variability.

**Baselines**. AFM is compared against several baseline deterministic and stochastic methods:

- **Regression**: A standard UNet is trained to predict high-res. output from low-res. input using MSE training. This serves as a purely deterministic approach, representing a baseline for direct super-resolution.
- **Conditional Diffusion Model (CDM)**: CDM maps Gaussian noise to the high-res. data while conditioned on the low-res. input. CDM does not explicitly model the deterministic component in the data.
- **Conditional Flow Matching (CFM)**: A variant of Flow Matching that interpolates between a Gaussian sample and a data sample, building deterministic mappings.
- **Corrective Diffusion Models (CorrDiff) (Mardani et al., 2023)**: A supervised regression is first trained to learn the mean $\mathbb{E}[\mathbf{x}|\mathbf{y}]$. The residual $e := \mathbf{x} - \mathbb{E}[\mathbf{x}|\mathbf{y}]$ is then used to train the diffusion model. Early stopping is used to mitigate the overfitting for the regression stage.

**Evaluation Metrics**. We report performance using standard metrics such as RMSE, Continuous Ranked Probability Score (CRPS), Mean Absolute Error (MAE), and Spread Skill Ratio (SSR). These metrics provide a comprehensive assessment of both the accuracy and uncertainty quantification of the model's predictions. RMSE, CRPS, and MAE measure the estimation error while SSR evaluates the model calibration. To calculate CRPS and SSR we

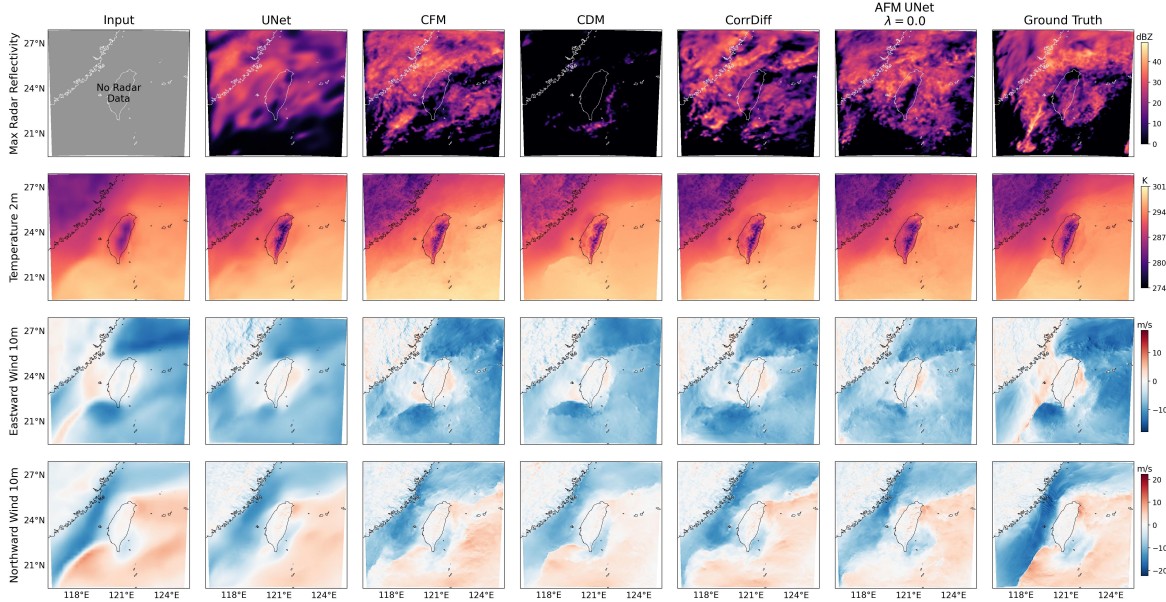

Figure 2: **AFM vs. baselines for different weather variables.** AFM generates more physically consistent outputs, while UNet (regression) output appears blurred, and CDM struggles to accurately reconstruct radar reflectivity. Note that radar reflectivity is not present in the input data and is entirely generated as a new channel.

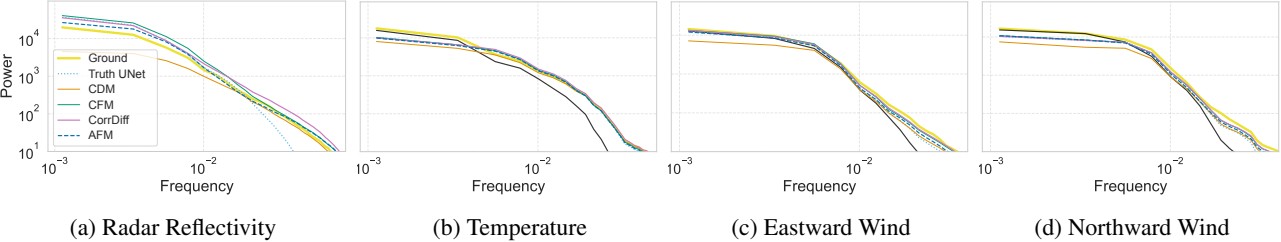

| (a) Radar Reflectivity | (b) Temperature | (c) Eastward Wind | (d) Northward Wind |

Figure 3: **AFM power spectra vs. baselines for CWA downscaling.** AFM exhibits superior spectral fidelity, closely aligning with the ground truth across all variables, with a particularly strong fidelity for the purely generated radar reflectivity. It consistently outperforms CorrDiff, especially in capturing high-frequency details across all variables.

produce 64 ensemble members using different seeds. These metrics are discussed in Appendix F.

**Network Architecture, Training, and Sampling**. For diffusion model training and sampling, we adopted EDM (Karras et al., 2022), a continuous-time diffusion model inspired by SDEs with effective auto-tuning (see Table 1 in (Karras et al., 2022)). We adopt most of the hyperparameters from EDM and make modifications as detailed in Appendix E. Note that EDM is used for CDM/CFM as well in a consistent manner.

### 5.1. Regional Downscaling for Taiwan

We focus on the task of super-resolving (downscaling) multiple weather variables for the Taiwan region, a challenging meteorological regime. The input coarse-resolution data at a 25 km scale comes from ERA5 (Hersbach et al., 2020), while the target fine-resolution 2 km scale data is sourced from the Central Weather Administration (CWA) (Central

Weather Administration (CWA), 2021). For evaluation, we use a common set of 205 randomly selected out-of-sample date and time combinations from 2021. Metrics and spectra are computed to compare AFM with baseline models. We utilize a 32-member ensemble; larger ensembles do not significantly alter the key findings. A detailed data description is provided first, followed by our observations.

**Dataset**. The dataset is derived from ERA5 reanalysis data (Hersbach et al., 2020), focusing on 12 variables including temperature, wind components, and geopotential height at two pressure levels, as well as surface-level variables e.g., 2-meter temperature and total column water vapor. The target output data (Central Weather Administration (CWA), 2021) covers a 900×900km region around Taiwan on a 448×448 grid. Hourly observations span four years (2018-2021), split into training (2018-2020) and evaluation (2021) sets. Input (ERA5) data is upsampled using bi-linear interpolation to match the target domain dimen-

Table 3: **AFM vs. baselines for Kolmogorov Flow for various misalignment degrees $\tau$.** AFM consistently demonstrates superior performance across varying levels of data misalignment, showcasing its robustness. While CDM exhibits greater variability, this comes at the expense of significantly reduced fidelity. Note that for deterministic models, CRPS is equivalent to MAE.

| $\tau$ | Metric | CFM | CDM | UNet | AFM |
|---|---|---|---|---|---|
| 3 | RMSE ↓ | 0.98 | 1.13 | 1.15 | **0.73** |
|  | CRPS ↓ | 0.52 | 0.58 | – | **0.37** |
|  | MAE ↓ | 0.69 | 0.80 | 0.82 | **0.51** |
|  | SSR → 1 | 0.54 | **0.69** | – | 0.62 |
| 5 | RMSE ↓ | 0.96 | 0.94 | 1.14 | **0.76** |
|  | CRPS ↓ | 0.52 | 0.48 | – | **0.40** |
|  | MAE ↓ | 0.69 | 0.67 | 0.82 | **0.54** |
|  | SSR → 1 | 0.58 | **0.70** | – | 0.58 |
| 10 | RMSE ↓ | 1.22 | 1.24 | 1.36 | **1.09** |
|  | CRPS ↓ | 0.67 | **0.65** | – | 0.65 |
|  | MAE ↓ | 0.89 | 0.89 | 1.00 | **0.77** |
|  | SSR → 1 | 0.56 | **0.76** | – | 0.23 |

Table 4: **Effect of adaptive $\sigma_{\mathbf{z}}$.** Experiments were conducted on the CWA $112 \times 112$ dataset with adaptive $\sigma_z$ and $\lambda = 0.0$. For both encoders adaptive $\sigma_{\mathbf{z}}$ significantly enhances performance across all variables and metrics.

| | Encoder Adapt. $\sigma_{\mathbf{z}}$ | $1 \times 1$ Conv. ✗ | $1 \times 1$ Conv. ✓ | UNet ✗ | UNet ✓ |
|---|---|---|---|---|---|
| **Radar** | RMSE ↓ | 5.01 | **4.82** | 5.06 | **4.95** |
|  | CRPS ↓ | 1.88 | **1.66** | 1.83 | **1.78** |
|  | SSR → 1 | 0.45 | **0.72** | 0.38 | **0.44** |
| **Temp.** | RMSE ↓ | 1.13 | **0.99** | 1.01 | **0.94** |
|  | CRPS ↓ | 0.66 | **0.59** | 0.57 | **0.54** |
|  | SSR → 1 | 0.39 | **0.47** | 0.36 | **0.40** |
| **E. Wind** | RMSE ↓ | 1.50 | **1.42** | 1.48 | **1.49** |
|  | CRPS ↓ | 0.86 | **0.78** | 0.85 | **0.85** |
|  | SSR → 1 | 0.48 | **0.68** | 0.46 | **0.53** |
| **N. Wind** | RMSE ↓ | 1.64 | **1.58** | 1.64 | **1.58** |
|  | CRPS ↓ | 0.94 | **0.89** | 0.93 | **0.95** |
|  | SSR → 1 | 0.47 | **0.66** | 0.46 | **0.50** |

sions (CWA) (Hu et al., 2019; Zhang et al., 2018). See Appendix G.1 for the detailed discussion of the datasets.

**Observations.** The performance of AFM is compared with various alternatives, and both deterministic and probabilistic skills are reported in Table 2. We examine three variants of AFM: with ($i$) small versus large encoders; ($ii$) with additional use of adaptive noise scaling; and ($iii$) conditioning in the large encoder limit. Notably, temperature is the most deterministic variable among the four listed, while radar is the most stochastic one.

Our main finding is that AFM consistently outperforms existing alternatives across different metrics for non-deterministic channels (radar and winds). For temperature, although AFM is not the top performer, one can tune $\lambda$ (larger value) in AFM to be as good as CorrDiff. This however compromises the stochastic prediction for non-deterministic channels; see the ablation in the Appendix section Table 12. Ablations are deferred to Appendix I due to space limitations.

Spectral analysis is crucial for assessing fidelity at different scales in weather prediction. Fig. 3 shows that AFM's spectra closely match the ground truth across variables. In contrast, the UNet-based regression scheme fails to generate high-frequency components. Interestingly, the conditional diffusion model (CDM), commonly used for image super-resolution, also lacks spectral fidelity. Regarding the calibration of the generated ensemble i.e., SSR, AFM provides the best balance, being closest to 1.0. While AFM does not acheive perfect calibration, it does improve the balance between the spread and RMSE skill of the generated ensemble, especially for surface wind channels.

## 5.2. Multiscale Kolmogorov-Flow

Our Multiscale Kolmogorov Flow dataset offers a simplified simulation of atmospheric dynamics, focusing on downscaling from a coarse to a fine grid while preserving physical structures. Kolmogorov flow (KF) is a well-known scenario where 2D fluid flow in a doubly periodic domain is forced by spatially varying source of momentum. To mimic the structure of the down-scaling problem we couple the KF flow ground truth to an otherwise unforced fluid system representing a coarse-resolution atmospheric simulation. The strength of this coupling $\tau$ controls how well the coarse simulation tracks the ground truth.

**Dataset**. The dataset is constructed by simulating dynamics governed by a system of partial differential equations involving vorticity fields $\zeta_l$ and $\zeta_h$, coupled through parameters like $\tau$ and influenced by steady-state forcing. The simulation uses a pseudo-spectral method on a $512 \times 512$ grid with a 3rd-order Adams-Bashforth time stepper. Different $\tau$ values (3, 5, 10) are used to generate training and test sets where higher values of $\tau$ looser coupling which translates to higher misalignment between low and high res simulations. Detailed descriptions of the equations and parameters are provided in Appendix G.2.

**Observations.** AFM consistently outperforms other methods across various skill metrics for different degrees of misalignment, denoted by $\tau$ (Table 3). Interestingly, while CDM appears to be the most calibrated, AFM demonstrates superior performance overall. In this case study, we use a $1 \times 1$ convolutional architecture for the AFM encoder. The ablations are deffered to Appendix K due to limited space. This advantage is further supported by the spectral analysis in Fig. 13, where AFM's spectra most closely align with the ground truth, highlighting its robustness in preserving physical structures even under significant misalignment.

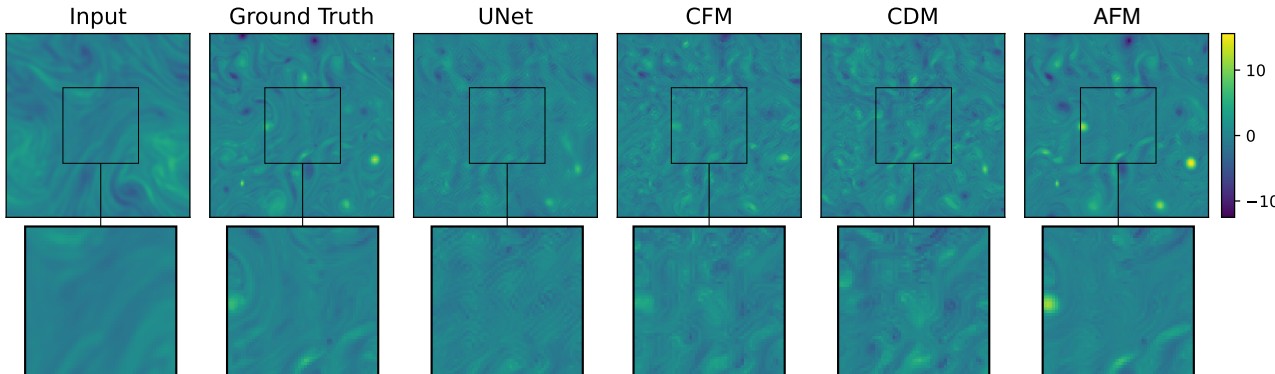

Figure 4: **AFM vs. baselines for Kolmogorov Flow and $\tau = 10$:** when the figures are zoomed in, it is apparent that AFM aligns closer to the ground truth, and the presence of high-frequency artifacts in the baseline models becomes more noticeable.

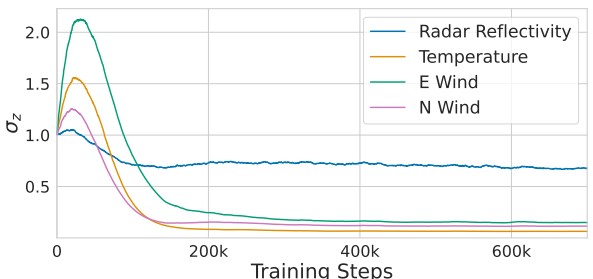

Figure 5: **Adaptive $\sigma_{\mathbf{z}}$ values over training steps for different channels.** The plot corresponds to AFM with $1 \times 1$ Conv. encoder and $\lambda = 0.25$. $\sigma_z$ increases early training due to high encoder error and subsequently converges as the encoder improves. Stochastic channels e.g., radar reflectivity exhibits the highest $\sigma$, indicating its stochastic nature, while temperature shows the lowest, reflecting its deterministic nature.

## 5.3. Ablations

**Role of noise scaling ($\sigma_{\mathbf{z}}$).** To investigate the effect of adaptive noise scaling, we tracked the per-channel noise scale during training (Figure 5) for a $1 \times 1$ Conv. encoder with $\lambda = 0.25$ on the CWA dataset. Initially set to 1, the noise scales grow in the early stages due to high encoder error, then stabilize and converge as training progresses and the encoder improves. Notably, radar reflectivity exhibits the highest $\sigma_z$ values throughout the training process, reflecting its stochastic nature. In contrast, the temperature channel consistently shows the lowest noise values, aligning with its more deterministic nature. These variations in noise across channels highlight the necessity of an adaptive noise scaling approach, particularly when compared to datasets (e.g., natural images) that exhibit more uniform noise statistics. To further validate this, we ablated the use of adaptive $\sigma_z$. As shown in Table 4, incorporating adaptive noise scaling significantly improves performance for both the $1 \times 1$ Conv and UNet encoders. This adaptability is crucial for handling misaligned data with varying degrees of stochasticity, ultimately enhancing the AFM model's overall performance and reliability in multiscale physics.

**Role of regularization $\lambda$.** We also investigated how the AFM model's performance depends on the $\lambda$ parameter, which balances encoder error and variability. For the simpler $1 \times 1$ Conv. encoder (with only 60 parameters), the model achieves its best accuracy when $\lambda = 0$, indicating this encoder is lightweight enough and no regularization is need for fitting the data. In contrast, the more complex UNet encoder benefits from moderate regularization ($\lambda = 0.25$), suggesting that overfitting is more likely with larger-capacity models and can be alleviated by incorporating some regularization. Overall, the choice of $\lambda$ should be informed by the complexity of the encoder: simpler networks require weaker to no regularization, while more expressive architectures can benefit from a modest regularization term. Further ablations and experiments can be found in the Appendix; see Appendices I and K.

## 6. Conclusions

We introduced Adaptive Flow Matching (AFM) to address misaligned data distributions in scientific super-resolution tasks (e.g., downscaling). AFM combines deterministic encoding of large-scale dynamics with Flow Matching, effectively capturing both deterministic and stochastic components of the data. Experiments on synthetic and real-world datasets demonstrated that AFM outperforms existing methods, especially when input and target distributions are significantly misaligned. Future avenues to explore include: ($i$) extending AFM to unpaired misaligned datasets and to other domains (e.g. MRI subsampling in medical imaging, natural images), ($ii$) theoretical analysis of its convergence, especially due to adaptive noise scaling, and ($iii$) leverage alternative stochastic encoders such as VAEs.

## Impact Statement

This work advances machine learning for scientific data, introducing a generative framework capable of resolving fine-scale physical phenomena from coarse, misaligned inputs. A primary application is in atmospheric downscal-

Table 5: **Effect of encoder regularization** $\lambda$. Experiments were conducted on the CWA $112 \times 112$ dataset with adaptive $\sigma_z$. Regularization with $\lambda = 0.25$ works best for UNet.

| Variable | Metric | $1 \times 1$ **Conv.** | | | **UNet** | | |
|---|---|---|---|---|---|---|---|
| | $\lambda$ | 0.00 | 0.25 | 1.00 | 0.00 | 0.25 | 1.00 |
| **Radar** | RMSE $\downarrow$ | **4.82** | 5.10 | 5.04 | 5.06 | **4.95** | 5.11 |
| | CRPS $\downarrow$ | **1.66** | 1.83 | 1.83 | 1.83 | **1.78** | 1.89 |
| | SSR $\rightarrow 1$ | **0.72** | 0.42 | 0.40 | 0.38 | **0.44** | 0.41 |
| **Temp.** | RMSE $\downarrow$ | 0.99 | **0.85** | 1.05 | 1.01 | **0.94** | 1.07 |
| | CRPS $\downarrow$ | 0.59 | **0.50** | 0.60 | 0.57 | **0.54** | 0.62 |
| | SSR $\rightarrow 1$ | **0.47** | 0.43 | 0.34 | 0.36 | **0.40** | 0.36 |
| **E. Wind** | RMSE $\downarrow$ | **1.42** | 1.46 | 1.50 | 1.48 | **1.49** | 1.53 |
| | CRPS $\downarrow$ | **0.78** | 0.84 | 0.88 | 0.85 | **0.85** | 0.91 |
| | SSR $\rightarrow 1$ | **0.68** | 0.46 | 0.44 | 0.46 | **0.53** | 0.41 |
| **N. Wind** | RMSE $\downarrow$ | **1.58** | 1.63 | 1.67 | 1.64 | **1.58** | 1.74 |
| | CRPS $\downarrow$ | **0.89** | 0.92 | 0.95 | 0.93 | **0.95** | 1.03 |
| | SSR $\rightarrow 1$ | **0.66** | 0.46 | 0.44 | 0.46 | **0.58** | 0.40 |

ing, where improved resolution and uncertainty quantification can enhance early detection of extreme weather events, contributing to disaster preparedness and public safety. We are not aware of direct negative societal impacts of this research.

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

# A. AFM versus CorrDiff

Here we illustrate experimentally how the two-stage workflow of CorrDiff relates to decreased diversity and under-calibration. We focus on the training dynamics of the UNet regression model, by training one and keeping various checkpoints (0.5M, 2M and 50M steps) along training. Then for each checkpoint we learn a diffusion model predict the residual. However, this approach is prone to overfitting in the first stage, limiting the overall performance. As shown in Table 6 when the regression model is trained for too many steps (e.g., 50M), the variability is significantly reduced. This reduction in SSR suggests that the regression component becomes overly confident, producing residuals that are narrowly concentrated around zero. Consequently the diffusion model only need to model limited variability, which diminishes its ability to generate diverse corrections at test-time. By contrast, an undertrained (0.5M steps) UNet leaves more stochasticity to be modeled, which leads to better-calibrated ensembles.

To further elucidate the issue we plot the training and validation losses of the encoder during training in Figure 6. As we can observe, the UNet regressor show a generalization gap that widens after approximately 500k training steps. This is evident from the divergence between the training and validation MSE losses across all channels. We also note that the radar and temperature channels have a wider generalization gap.

These observations highlight a critical limitation of CorrDiff's two-stage methodology: the initial regression stage is prone to overfitting, which in turn constrains the diffusion model's capacity to generalize effectively. To address this issue, the authors of CorrDiff use early-stopping to chose a less overfit UNet.

Our proposed AFM method addresses this limitation by adopting an end-to-end training setup. AFM simultaneously balances deterministic and stochastic objectives, preserving sufficient variability in the signal, reducing the risk of overfitting while at the same time produces more accurate results. An important part of our method is that it takes into account the different stochasticity levels between channels (see Figure 9 using the adaptive noise scaling, and overall mitigates the overfitting issue in a more principled and effective way.

To gather the above results for CorrDiff we use the same UNet from the CorrDiff paper which corresponds to the 89M-parameter model (XL in Table 7). This makes the RMSE and CRPS slightly different to the rest of our experiments where we used 12M-parameter UNet (L).

Table 6: **Effect of two-stage training on ensemble diversity.** This table shows how the number of training steps for the UNet regressor in the first stage influences the final performance of the entire CorrDiff pipeline. In the second stage, different diffusion models are trained using UNet checkpoints obtained after 0.5M, 2M, and 50M steps. Results indicate that using less-trained UNet models yields better calibration and higher stochastic variability (SSR), while heavily-trained UNet models reduce ensemble diversity. This suggests that once a UNet becomes too specialized or biased, the second-stage diffusion model struggles to correct it and maintain variability in its outputs.

| Variable | Metric | U-Net Training steps | | |
| --- | --- | --- | --- | --- |
| | | 0.5M | 2M | 50M |
| | RMSE $\downarrow$ | 5.08 | 5.28 | 5.13 |
| Radar | CRPS $\downarrow$ | 1.81 | 1.89 | 2.10 |
| | SSR $\rightarrow 1$ | 0.52 | 0.35 | 0.14 |
| | RMSE $\downarrow$ | 0.96 | 0.93 | 0.91 |
| Temp. | CRPS $\downarrow$ | 0.57 | 0.58 | 0.53 |
| | SSR $\rightarrow 1$ | 0.41 | 0.28 | 0.31 |
| | RMSE $\downarrow$ | 1.51 | 1.53 | 1.48 |
| E. Wind | CRPS $\downarrow$ | 0.83 | 0.89 | 0.87 |
| | SSR $\rightarrow 1$ | 0.60 | 0.42 | 0.39 |
| | RMSE $\downarrow$ | 1.72 | 1.70 | 1.65 |
| N. Wind | CRPS $\downarrow$ | 0.97 | 1.01 | 0.99 |
| | SSR $\rightarrow 1$ | 0.56 | 0.39 | 0.37 |

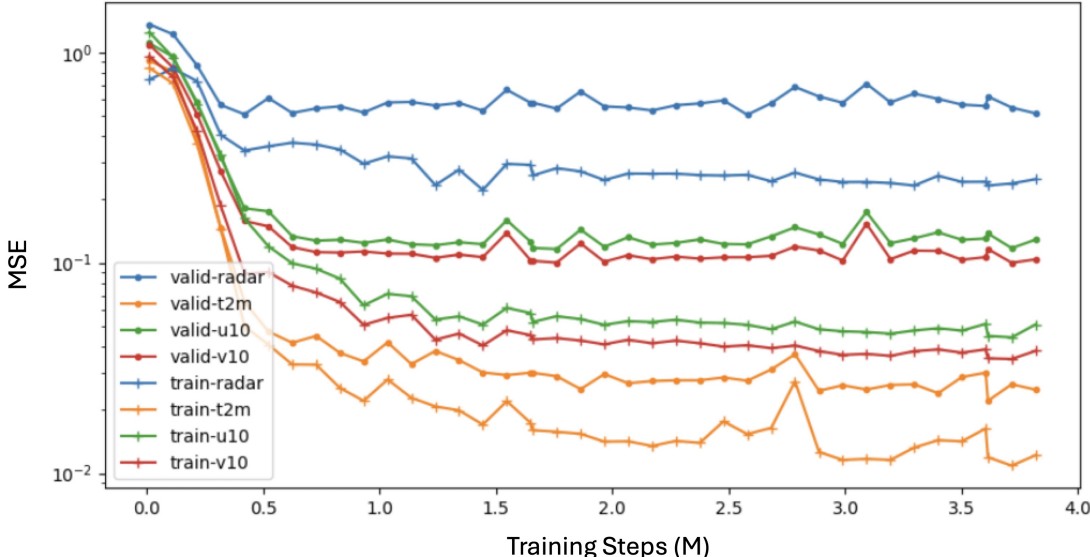

Figure 6: **UNet train and validation loss during training on CWA** $448 \times 448$ **data.** Evolution of training and validation MSE for the UNet regressor across training steps. We observe that the UNet starts overfitting as early as 500k steps. Furthermore, certain variables like radar reflectivity(blue) and temperature(yellow) show higher amounts of overfitting. This behavior is the reason that two-stage approaches like CorrDiff utilize early stopping to avoid overfitting the training data, because this is very difficult to correct on the second stage. AFM resolves this issue by leveraging end-to-end training determinsticn loss weghting as well as adaptive noise scaling.

## B. Proof of Proposition 1

**Proposition 1**. *For the perturbation model $\mathbf{x}_t = \mathbf{x} + \sigma_t \boldsymbol{e} + \sigma_t \boldsymbol{\epsilon}$, where the noise standard deviation is given by $\sigma_t := (1-t)\sigma_z$, the residual error by $\boldsymbol{e} := (\mathcal{E}(\mathbf{y}) - \mathbf{x})/\sigma_z$, and the noise $\boldsymbol{\epsilon} \sim \mathcal{N}(0,1)$, the Flow Matching for joint training of the encoder and flow reduces to the denoising objective:*

$$\min_{\mathcal{E}, \boldsymbol{\theta}} \mathbb{E}_{\mathbf{x}, \mathbf{y}, \sigma_t \sim \mathcal{U}[0, \sigma_z]} \left[ (\sigma_z/\sigma_t)^2 \left\| \mathcal{D}_{\boldsymbol{\theta}}(\mathbf{x}_t, \sigma_t) - \mathbf{x} \right\|^2 \right]. \tag{11}$$

**Proof**. Consider the linear interpolant connecting $\mathbf{z} \sim \mathcal{N}(\mathcal{E}(\mathbf{y}), \sigma_z \mathbf{I})$ to the target distribution $p(\mathbf{x})$. This process can be expressed as:

$$\mathbf{x}_t = (1-t)(\mathcal{E}(\mathbf{y}) + \sigma_z \boldsymbol{\epsilon}) + t\mathbf{x} \tag{12}$$
$$= (1-t)\mathcal{E}(\mathbf{y}) + t\mathbf{x} + (1-t)\sigma_z \boldsymbol{\epsilon} \tag{13}$$
$$= (1-t)(\mathcal{E}(\mathbf{y}) - \mathbf{x}) + \mathbf{x} + (1-t)\sigma_z \boldsymbol{\epsilon} \tag{14}$$
$$= \mathbf{x} + (1-t)\sigma_z(\boldsymbol{e} + \boldsymbol{\epsilon}), \tag{15}$$

where $\boldsymbol{e} = (\mathcal{E}(\mathbf{y}) - \mathbf{x})/\sigma_z$.

Define $\sigma_t = (1-t)\sigma_z$ and reinterpret time $t$ in terms of the noise level $\sigma_t$, as is typical in continuous noise sampling (e.g., EDM). The perturbation kernel then becomes $\mathbf{x}_t = \mathbf{x} + \sigma_t(\boldsymbol{e} + \boldsymbol{\epsilon})$.

Now, consider the training objective in (4):

$$\min_{\mathcal{E}, \boldsymbol{\theta}} \mathbb{E}_{t, \mathbf{x}, \mathbf{z} \sim \mathcal{N}(\mathcal{E}(\mathbf{y}), \sigma_z)} \left[ \left\| \boldsymbol{\nu}_\theta(\mathbf{x}_t, t) - (\mathbf{x} - \mathbf{z}) \right\|^2 \right].$$

where the velocity field expressed as a denoiser (or equivalently score form (Vincent, 2011)):

$$\boldsymbol{\nu}_{\boldsymbol{\theta}}(\mathbf{x}_t, t) = \frac{\mathcal{D}_{\boldsymbol{\theta}}(\mathbf{x}_t, \sigma_t) - \mathbf{x}_t}{1 - t}, \tag{16}$$

which is the reverse-time version of Equation (20), since in Flow Matching, time progresses from $t = 0$ to $t = 1$. Substituting $\boldsymbol{\nu}_{\boldsymbol{\theta}}(\mathbf{x}_t, t)$ into the Flow Matching objective (4) and simplifying using $\mathbf{x}_t = (1-t)\mathbf{z} + t\mathbf{x}$ and $\sigma_t/\sigma_z = 1 - t$, we obtain the denoising objective in (11), completing the proof.

# C. Flow Matching and Diffusion Models

## C.1. Diffusion Models (DM)

Diffusion models generate data by transforming a base distribution, often Gaussian noise, into the target data distribution $p_0(\mathbf{x})$. In the forward diffusion process, Gaussian noise with standard deviation $\sigma$ is added to the data, producing a sequence of distributions $p_0(\mathbf{x}; \sigma)$. As $\sigma$ increases, the data distribution approaches pure noise. The backward process then denoises samples, starting from noise drawn from $\mathcal{N}(0, \sigma_{\max}^2 \mathbf{I})$ and progressively reducing the noise to recover the data distribution.

Considering the variance-exploding elucidated diffusion model (EDM), both the forward and backward processes are described by SDEs. The forward SDE is:

$$d\mathbf{x}_t = \sqrt{2\dot{\sigma}_t \sigma_t}\, d\boldsymbol{\omega}_t, \tag{17}$$

while the backward SDE is:

$$d\mathbf{x}_t = -2\dot{\sigma}_t \sigma_t \nabla_{\mathbf{x}_t} \log p(\mathbf{x}_t; \sigma_t)\, dt + \sqrt{2\dot{\sigma}_t \sigma_t}\, d\boldsymbol{\omega}_t. \tag{18}$$

In EDM, denoising score matching is used to learn the score function $\nabla_{\mathbf{x}} \log p(\mathbf{x}; \sigma)$, essential for the reverse diffusion process. A denoising neural network $D_\theta(\mathbf{x}; \sigma)$ is trained as:

$$\min_\theta \mathbb{E}_{\mathbf{x} \sim p_0, \sigma_t \sim p_\sigma, \mathbf{n} \sim \mathcal{N}(0, \sigma_t^2 \mathbf{I})} \left[ \| \mathcal{D}_{\boldsymbol{\theta}}(\mathbf{x} + \mathbf{n}, \sigma_t) - \mathbf{x} \|^2 \right], \tag{19}$$

and the score function is constructed via $\nabla_{\mathbf{x}_t} \log p(\mathbf{x}_t; \sigma_t) = (\mathcal{D}_{\boldsymbol{\theta}}(\mathbf{x}_t, \sigma_t) - \mathbf{x})/\sigma_t^2$.

For completeness, we note that the ODE version of this backward process (used in the main text as Eq. 20) directly links EDM to Flow Matching-based approaches.

## C.2. Connections Between Flow Matching and EDM

While both EDM and Flow Matching methods aim to transform distributions, Flow Matching does so using ODEs for deterministic evolution, whereas elucidated diffusion models (EDM) leverage SDEs for stochastic denoising. An ODE formulation for diffusion models bridges the two methods, allowing Flow Matching to benefit from formulations and network parameterizations introduced for diffusion models (Karras et al., 2022; Song et al., 2021). For the simple noise schedule $\sigma_t = \sqrt{t}$, the ODE for continuous-time EDM is:

$$\frac{d\mathbf{x}_t}{dt} = \frac{\mathbf{x}_t - \mathcal{D}_{\boldsymbol{\theta}}(\mathbf{x}_t, \sigma_t)}{t}, \tag{20}$$

where the right-hand side acts as the velocity field, linking diffusion dynamics to Flow Matching. Note that the diffusion process runs backward in time from $t = 1$ to $t = 0$, whereas the Flow Matching process proceeds in the opposite direction.

# D. Connection of AFM to Residual Learning

Consider the case where we have a pre-trained encoder $\mathcal{E}$, which has been trained using a supervised regression loss, for instance. In this scenario, starting from the forward diffusion process in Equation (5), if we subtract both sides by the encoder output $\mathcal{E}(\mathbf{y})$ and define the residual error as $\boldsymbol{e}_t := \mathbf{x}_t - \mathcal{E}(\mathbf{y})$, we arrive at a form that closely resembles a standard Flow Matching forward process with Gaussian noise as the base distribution. This is expressed as:

$$\boldsymbol{e}_t = t\boldsymbol{e} + (1-t)\boldsymbol{\epsilon}, \qquad t \in [0, 1], \tag{21}$$

where $\boldsymbol{e}$ represents the residual error between the target $\mathbf{x}$ and the encoder output $\mathcal{E}(\mathbf{y})$, and $\boldsymbol{\epsilon} \sim \mathcal{N}(0, 1)$ is the noise.

This simple process facilitates the construction of a backward process, where one can learn the velocity field by minimizing the Flow Matching loss in Equation (2). This approach closely mirrors the CorrDiff method proposed in (Mardani et al., 2023), which leverages residual learning to train diffusion models. CorrDiff has demonstrated considerable success in capturing small-scale details in generative tasks, particularly where precise reconstruction of fine structures is required.

However, as discussed in Section 2, the initial supervised training of the encoder often leads to near-perfect matching between $\mathcal{E}(\mathbf{y})$ and $\mathbf{x}$. While this may seem desirable, it can result in overfitting and reduced model calibration, as also evidenced by the results in Table 6 and discussed in Appendix A. Therefore, balancing this residual learning approach with appropriate regularization is critical to maintaining the model's ability to generalize effectively to unseen data.

Table 7: Details of different encoder sizes used in the experiments. For the UNet the channel multipliers are applied to the base channel size of 32 at each layer.

| Encoder | Channel Multipliers | Number of Parameters |
|---|---|---|
| XL (only used in Appendix A) | [1, 4, 4, 8, 8] | 89M |
| L | [1, 2, 2, 4, 4] | 12M |
| M | [1, 2, 2, 2, 2] | 5M |
| S | [1, 1, 2, 2, 2] | 1M |
| XS | [1, 1, 1, 2, 2] | 0.2M |
| $1 \times 1$ Conv. | - | 60 |

## E. Network Architecture and Training

For diffusion model training and sampling, we use EDM (Karras et al., 2022), a continuous-time diffusion model available with a public codebase. EDM provides a physics-inspired design based on ODEs, auto-tuned for our scenario (see Table 1 in (Karras et al., 2022)). We adopt most of the hyperparameters from EDM and make modifications as listed below.

**Architecture of Diffusion Model**: To cover the large field-of-view $448 \times 448$, we adapt the UNet from (Song & Ermon, 2019) by expanding it to 5 encoder and 5 decoder layers. The base channel size is 32, multiplied by [1, 2, 2, 4, 4] across layers. Please note this is a scaled down version of the model used in (Mardani et al., 2023) due to computational constrains. Attention resolution is set to 28. Time representation is handled via positional embedding, though this is disabled in the regression network, as no probability flow ODE is involved. No data augmentation is applied. The UNet has 12 million parameters, and we add 4 channels for sinusoidal positional embedding to improve spatial consistency, following practices in (Dosovitskiy, 2020; Carion et al., 2020).

**Architecture of Encoder Model** For the encoder $\mathcal{E}$, we use two architectures: (1) a simple $1 \times 1$ convolution layer, and (2) a UNet similar to the diffusion UNet but without time embedding. For the UNet encoder we ablate various parameter counts from 0.2M to 12M, architectural details of these can be found in Table 7 and the results of the albations in Appendix I. For the experiment where only the UNet was used for regression, the model has 12M parameters (L). For CorrDiff then same 12M model was used for the regression network and the diffusion model. For the results in CorrDiff under-calibration (Appendix A) we used the same model as in (Mardani et al., 2023) with 89M parameters (XL).

**Optimizer**: We use the Adam optimizer with a learning rate of $10^{-4}$, $\beta_1 = 0.9$, $\beta_2 = 0.99$. Dropout is applied with a rate of 0.13. Hyperparameters follow the guidelines in EDM (Karras et al., 2022).

**Noise Schedule**: For AFM and CFM, we use a continuous noise schedule sampled uniformly $\sigma \in \mathcal{U}[0, \sigma_z]$. For CDM and CorrDiff, we use EDM's optimized log-normal noise schedule, $\sigma \sim \mathrm{lognormal}(-1.2, 1.2)$.

**Training**: The regression network receives 12 input channels from the ERA5 data, while diffusion training concatenates these 12 input channels with 4 noise channels. EDM randomly selects noise variance aiming to denoise samples per mini-batch. CFM, CDM, CFM and CorrDiff are trained for 50 million steps, whereas the regression UNet is trained for 20 million steps. For AFM we evaluate the encoder's RMSE every 10k steps and update $\sigma_z$ using EMA with $\alpha = 0.9$.Training is distributed across 8 DGX nodes, each with 8 A100 GPUs, using data parallelism and a total batch size of 512.

**Conditioning**: AFM starts from the encoder output and learns the stochastic dynamics directly in the latent space. CDM and CFM map Gaussian noise to the high-resolution output space while being conditioned on the low-resolution input. Conditioning works by concatenating the low-resolution input with the noise, as described in Batzolis et al. (2021) and Saharia et al. (2022). AFM can also be conditioned on the low-resolution input, the effect of which we explore in the ablations in Appendix I.

**Sampling**: Our sampling process employs Euler integration with 50 steps across all methods. We begin with a maximum noise variance $\sigma_{\max}$ and decrease it to a minimum of $\sigma_{\min} = 0.002$. The value of $\sigma_{\max}$ varies depending on the method: for CDM and CorrDiff, we use $\sigma_{\max} = 800$, as per the original implementation in (Mardani et al., 2023); for CFM, we set $\sigma_{\max} = 1$, as specified in (Lipman et al., 2022); and for AFM, we use the $\sigma_z$ value learned during training.

## F. Evaluation Metrics

### F.1. RMSE

The Root Mean Square Error (RMSE) is a standard evaluation metric used to measure the difference between the predicted values and the true values (Chai & Draxler, 2014). In the context of our problem, let $\mathbf{x}$ be the true target and $\hat{\mathbf{x}}$ be the predicted value. The RMSE is defined as:

$$\text{RMSE} = \sqrt{\mathbb{E}\left[\|\mathbf{x} - \hat{\mathbf{x}}\|^2\right]}. \tag{22}$$

This metric captures the average magnitude of the residuals, i.e., the difference between the predicted and true values. A lower RMSE indicates better model performance, as it suggests the predicted values are closer to the true values on average. RMSE is sensitive to large errors, making it an ideal choice for evaluating models where minimizing large deviations is critical.

### F.2. CRPS

The Continuous Ranked Probability Score (CRPS) is a measure used to evaluate probabilistic predictions (Wilks, 2011). It compares the entire predicted distribution $F(\hat{\mathbf{x}})$ with the observed data point $\mathbf{x}$. For a probabilistic forecast with cumulative distribution function (CDF) $F$, and the true value $\mathbf{x}$, the CRPS is given by:

$$\text{CRPS}(F, \mathbf{x}) = \int_{-\infty}^{\infty} \left(F(y) - \mathbb{I}(y \geq \mathbf{x})\right)^2 dy, \tag{23}$$

where $\mathbb{I}(\cdot)$ is the indicator function. Unlike RMSE, CRPS provides a more comprehensive evaluation of both the location and spread of the predicted distribution. A lower CRPS indicates a better match between the forecast distribution and the observed data. It is especially useful for probabilistic models that output a distribution rather than a single point prediction.

When applying CRPS to a finite ensemble of size $m$ approximating $F$ with the empirical CDF incurs an $O(1/m)$ bias favoring models with less spread. For small $m$ unbiased versions of the formulas should be used instead (Zamo & Naveau, 2018), but for the ensemble sizes here this is a small effect, so we used the more common biased formulas.

### F.3. Spread Skill Ratio

The Spread-Skill Ratio (SSR) evaluates the reliability of the predicted uncertainty by comparing the spread (variance) of the predicted distribution with the accuracy of the predictions (Gneiting & Raftery, 2004). Let $\sigma_{\hat{\mathbf{x}}}$ be the standard deviation of the predicted distribution and RMSE as defined above. The SSR is defined as:

$$\text{SSR} = \frac{\sigma_{\hat{\mathbf{x}}}}{\text{RMSE}}. \tag{24}$$

An SSR value close to 1 indicates that the predicted uncertainty (spread) is well-calibrated with the model's predictive skill. If the SSR is less than 1, the model underestimates uncertainty, while an SSR greater than 1 indicates that the model overestimates uncertainty. This metric is particularly useful in evaluating the quality of probabilistic forecasts in terms of their sharpness (spread) and accuracy (skill).

## G. Details of the Datasets

Further details and visualizations of the ERA5-CWA and KF datasets, used throughout the paper, are presented here.

### G.1. ERA5-CWA Dataset

Table Table 8 summarizes the input-output channels and the corresponding resolutions. It is evident that the input and output channels generally differ, and even those that do overlap, such as (Temperature, East Wind, North Wind), are not perfectly aligned. For instance, comparing the Eastward Wind (10m) in the contour plots reveals the eye of the typhoon located northeast of the Taiwan region; see Fig. 7. Notably, the typhoon's eye shifts in the output due to the datasets

Table 8: **ERA5-CWA Variables**: Input and target variables for the ERA5 to CWA downscaling task include both single-level and pressure-level variables, the latter at 850 and 500 hPa.

| Description | Input | Output |
|---|---|---|
| Pixel Size | $36 \times 36$ | $448 \times 448$ |
| Single-Level Channels | Total Column Water Vapor
Temperature at 2 Meters
East Wind at 10 Meters
North Wind at 10 Meters | Maximum Radar Reflectivity
Temperature at 2 Meters
East Wind at 10 Meters
North Wind at 10 Meters |
| Pressure-Level Channels | Temperature
Geopotential
East Wind
North Wind | -
-
-
- |

originating from two different simulations, which solve distinct sets of partial differential equations (PDEs) at significantly different resolutions, resulting in divergent trajectories.

**Input Data (ERA5)**. This data for this study are derived from the ERA5 reanalysis, which provides a comprehensive set of atmospheric variables at various vertical levels (Hersbach et al., 2020). For our analysis, we selected a subset of 12 variables. These include four variables (temperature, East and North components of the horizontal wind vector, and geopotential height) at two pressure levels (500 hPa and 850 hPa). Additionally, we incorporated single-level variables: 2-meter temperature, 10-meter wind vector components, and total column water vapor.

**Target Data (CWA)**. The horizontal range of these data encompasses a $900 \times 900$-km region containing Taiwan, with a horizontal resolution of approximately many output variables. We focus on four variables, three common to the input data – surface temperature and ruface horizontal wind components – and one of which is distinctly related to precipitating hydrometeors, the composite synthetic radar reflectivity at time of data assimilation. We represent the high-dimensional target data as $\mathbf{x} \in \mathbb{R}^{H \times W \times C}$, where $H = W = 448$ and $C = 4$. See table 8.

The dataset encompasses approximately four years (2018-2021) of observations, sampled at hourly intervals. For model development and validation, the data were partitioned chronologically. The training set comprises observations from 2018 to 2020, totaling 24,601 data points. The remaining data from 2021, consisting of 6,501 data points, were reserved for evaluation purposes. This temporal split allows for an assessment of the model's performance on future, unseen data, simulating real-world application scenarios.

Lastly, we upsample the input data to a $448 \times 448$ grid using bilinear interpolation to match the output resolution, a common practice with residual networks for consistency (Hu et al., 2019; Zhang et al., 2018) (see Figure 7b for an example of input vs. target misalignment).

### G.2. Kolmogorov-Flow Dataset

A representative input-output sample of the KF dataset is shown in Fig. 8 for different misalignment degrees $\tau$.

**Dataset description**. We construct a toy dataset by simulating the dynamics given by:

$$
\begin{aligned}
\zeta_h + J(\psi_h, \zeta_h) &= F + \nu_h \nabla^7 \zeta_l - \zeta_l \tau_r^{-1} \\
\zeta_l + J(\psi_l, \zeta_l) &= -\tau^{-1}(\zeta_l - \zeta_h) + \nu_l \nabla^7 \zeta_l - \zeta_l \tau_r^{-1} \\
\nabla^2 \psi_l &= \zeta_l \\
\nabla^2 \psi_h &= \zeta_h.
\end{aligned}
\tag{25}
$$

Here, $J(f, g) = f_x g_y - f_y g_x$ is the Jacobian operator. The stream function is related to the velocity field by $\nabla \psi = (-u, v)$, implying that $\nabla \psi \cdot (u, v) = 0$, so that velocity points along contours of the stream-function. $\zeta_{l,h}$ represents the vorticity.

The $\zeta_l$ field represents a coarse-resolution simulation nudged towards a high-resolution $\zeta_h$. The parameter $\tau$ controls the coupling strength between the $\zeta_l$ and $\zeta_h$ fields. A steady-state forcing $F = 10 \cos(10x)$ injects energy into the small-scale field $\zeta_h$ but not the low-resolution $\zeta_l$, mimicking the injection of energy by sub-grid processes like convection or flow over topography. Stronger dissipation $\nu_l \gg \nu_h$ is used to limit the effective resolution of the large-scale field. A small amount

of Rayleigh damping $\tau_r = 100$ is added to limit the pile-up of energy at large-scales.

These equations are solved using a standard pseudo-spectral method on the GPU. The 3rd-order Adams-Bashforth time stepper is used for all but the hyper-viscosity terms; for these stiff terms, we use an backward Euler time stepper. The resolution is $512 \times 512$ and the timestep $dt = 0.001$. A 2/3 de-aliasing filter is applied in spectral space every timestep (Orszag, 1971). Outputs are saved every $\delta = 0.2$ time units.

We create datasets for different $\tau$ values: $3, 5, 10, 20$. Higher $\tau$ corresponds to greater misalignment between the coarse and high-resolution simulations. This variation allows us to assess the robustness of our method across different levels of coupling and identify potential thresholds in $\tau$ beyond which certain downscaling approaches may become unreliable. For each $\tau$ value, we generate a dataset comprising $100,000$ training points and $500$ test points.

### G.3. Misalignment

A key challenge in both datasets is the misalignment between input and output fields, arising from differences in the PDEs used at each scale. These discrepancies lead to both large-scale and fine-scale shifts.
In the ERA5-CWA dataset (Figure 7b), for example, storm centers in the eastward and northward wind fields are misaligned by several grid points— etween the input and target. An additional difficulty in this dataset is that the radar reflectivity channel is only present in the target and must be entirely reconstructed from the other input variables.
The Kolmogorov-Flow dataset was specifically designed to study varying degrees of misalignment. Here, the degree of mismatch is controlled by the coupling parameter $\tau$. For smaller values (e.g., $\tau = 3$), the input and target fields remain largely aligned, with only minor local discrepancies and some loss of fine-scale detail. As $\tau$ increases (e.g., $\tau = 10$), the misalignment becomes more pronounced, introducing clear positional shifts between corresponding features, as shown in Figure 8. This setup provides a controlled way to assess how different levels of misalignment affect model performance.

## H. Analysis of Adaptive $\sigma_{\mathbf{z}}$ during Training

To evaluate the behavior of the adaptive noise scaling mechanism in our Adaptive Flow Matching (AFM) model, we monitored the sigma values across different channels during the training process. Results depicted in Figure 9 correspond to the model with a 1x1 conv. encoder and $\lambda = 0.25$ trained on the CWA dataset. The sigma values are initially set to 1 for all channels. During the early stages of training, sigma increases across the channels, due to the high encoder error. As training progresses and the encoder's performance improves, the sigma values begin to stabilize and converge towards their final values.

Notably, the radar reflectivity channel exhibits the highest sigma values throughout the training process, reflecting its inherently stochastic nature. This is consistent with our understanding that radar data contains significant variability and uncertainty. In contrast, the temperature channel consistently shows the lowest sigma values, aligning with its more deterministic characteristics. These variations in sigma across channels underscore the effectiveness of our adaptive noise scaling approach, as it allows the model to appropriately adjust noise levels based on the inherent uncertainty of each channel. This adaptability is crucial for managing misaligned data with differing degrees of stochasticity, thereby enhancing the overall performance and reliability of the AFM model in multiscale physics applications.

## I. CWA Ablation Studies

To evaluate the effectiveness of the AFM model and understand the impact of its components, we conducted ablation studies on the CWA weather downscaling task at $112 \times 112$ resolution. We focused on varying the $\lambda$ parameter, different encoder types, the use of adaptive $\sigma_{\mathbf{z}}$, and $\mathbf{y}$ conditioning. The results are summarized in Tables 9, 10, and 11.

### I.1. Effect of $\lambda$ Parameter and Encoder Type

Table 9 presents the performance of the AFM model with different $\lambda$ values and encoder types. The $\lambda$ parameter controls the trade-off between data fitting and uncertainty regularization in the AFM model.

$1 \times 1$ **Conv. Encoder:** For the $1 \times 1$ Conv. encoder, setting $\lambda = 0$ achieves the best performance across most variables, particularly for radar reflectivity, eastward wind, and northward wind. This indicates that the model benefits from minimal regularization when using a simpler encoder, allowing it to focus on fitting the data closely. The low parameter count (only 60 parameters) of the $1 \times 1$ Conv. encoder might limit its ability to capture all the information in the low-resolution input,

making less regularization advantageous.

**UNet Encoder:** In contrast, the UNet encoder shows improved performance with a small regularization parameter of $\lambda = 0.25$. This suggests that the more complex architecture of the UNet benefits from some regularization that prevents overfitting and leads to better accuracy and uncertainty estimates across multiple variables.

### I.2. Impact of adaptive $\sigma_{\mathbf{z}}$

Table 10 examines the effect of enabling or disabling adaptive $\sigma_{\mathbf{z}}$ for both encoder types.

**Findings:** Enabling adaptive $\sigma_{\mathbf{z}}$ consistently enhances the model's performance across all variables and both encoder types. The improvements in RMSE, CRPS, and SSR metrics suggest that adaptive $\sigma_{\mathbf{z}}$ allows the model to better capture the underlying uncertainty in the data. This adaptive approach provides the flexibility to model variable levels of uncertainty across different regions and variables, leading to more accurate and reliable predictions.

### I.3. Effect of y Conditioning

Table 11 assesses the impact of enabling or disabling $\mathbf{y}$ conditioning in the AFM model for both encoders.

$1 \times 1$ **Conv. Encoder:** Disabling $\mathbf{y}$ conditioning provides better results across most metrics for this encoder. Given the simplicity of the $1 \times 1$ Conv. encoder and its limited parameter count, it may not effectively utilize the additional information provided by $\mathbf{y}$ conditioning. The model performs better when focusing on directly mapping the input to the output without the added complexity.

**UNet Encoder:** Enabling $\mathbf{y}$ conditioning yields the best results for the UNet encoder. The more complex architecture of the UNet can leverage the additional context from $\mathbf{y}$ conditioning to improve its predictions. This demonstrates the capacity of the UNet encoder to capture and utilize supplementary information, enhancing both accuracy and uncertainty estimation.

### I.4. Summary

These ablation studies highlight the strengths of the AFM model and its components in weather downscaling tasks:

- **Effectiveness of Adaptive $\sigma_{\mathbf{z}}$:** Enabling adaptive $\sigma_{\mathbf{z}}$ consistently improves model performance across both encoder types and all variables. This underscores the importance of modeling spatially varying uncertainty in complex weather data.

- **Encoder Choice and Regularization:** The $1 \times 1$ Conv. encoder performs best without regularization ($\lambda = 0$), indicating that minimal regularization benefits simpler models. For the UNet encoder, a small regularization parameter ($\lambda = 0.25$) yields better results, suggesting that regularization helps prevent overfitting in more complex models.

- **Impact of y Conditioning:** $\mathbf{y}$ conditioning enhances performance for the UNet encoder but not for the $1 \times 1$ Conv. encoder. This suggests that the effectiveness of incorporating additional context depends on the model's capacity to utilize that information.

Overall, the AFM model demonstrates strong performance and flexibility in weather downscaling tasks. By carefully selecting model components such as encoder type, adaptive $\sigma_{\mathbf{z}}$, and regularization parameter $\lambda$, the AFM can be tailored to balance predictive accuracy and uncertainty estimation effectively. These findings highlight the potential of the AFM model as a powerful tool for probabilistic weather downscaling.

## J. CWA Ensemble Analysis

For a few representative samples, several ensemble members and the ensemble mean are shown in **??** 11a–J. The generated samples using different random seeds exhibit notable diversity, particularly for channels like Radar Reflectivity. This diversity confirms the model's ability to produce a well-dispersed ensemble, which is crucial for achieving a calibrated and reliable probabilistic forecast. Additionally, the ensemble mean closely aligns with the true target, indicating that the model successfully captures the underlying physical processes while preserving uncertainty across channels.

Animated PNGs of ensemble members for different models and $\tau$ are provided at `https://t.ly/ZCq9Z`.

Table 9: **Encoder and $\lambda$ Ablations on CWA** $112 \times 112$ **Dataset.** For this table we keep the best configurations for each $\lambda$ value based on radar reflectivity. We separate per $1 \times 1$ Conv. and UNet encoder to elucidate differences. Best results for each encoder are highlighted in bold. No regularization ($\lambda = 0$) works best for the $1 \times 1$ Conv. encoder, while for UNet, a small $\lambda = 0.25$ value yields the best results across multiple variables. Overall the $1 \times 1$ Conv. without regularization is the best configuration.

| Encoder | $\lambda$ | Radar | | | Temperature | | | Eastward Wind | | | Northward Wind | | |
|---|---|---|---|---|---|---|---|---|---|---|---|---|---|
| | | RMSE ↓ | CRPS ↓ | SSR → 1 | RMSE ↓ | CRPS ↓ | SSR → 1 | RMSE ↓ | CRPS ↓ | SSR → 1 | RMSE ↓ | CRPS ↓ | SSR → 1 |
| $1 \times 1$ Conv. | 0.00 | **4.82** | **1.66** | **0.72** | 0.99 | 0.59 | **0.47** | **1.42** | **0.78** | **0.68** | **1.58** | **0.89** | **0.66** |
| | 0.25 | 5.10 | 1.83 | 0.42 | **0.85** | **0.50** | 0.43 | 1.46 | 0.84 | 0.46 | 1.63 | 0.92 | 0.46 |
| | 1.00 | 5.04 | 1.83 | 0.40 | 1.05 | 0.60 | 0.34 | 1.50 | 0.88 | 0.44 | 1.67 | 0.95 | 0.44 |
| UNet | 0.00 | 5.06 | 1.83 | 0.38 | 1.01 | 0.57 | 0.36 | 1.48 | 0.85 | 0.46 | 1.64 | 0.93 | 0.46 |
| | 0.25 | **4.95** | **1.78** | **0.44** | **0.94** | **0.54** | **0.40** | **1.49** | **0.85** | **0.53** | **1.58** | **0.95** | **0.50** |
| | 1.00 | 5.11 | 1.89 | 0.41 | 1.07 | 0.62 | 0.36 | 1.53 | 0.91 | 0.41 | 1.74 | 1.03 | 0.40 |

Table 10: **Adaptive $\sigma_{\mathbf{z}}$ Ablation on CWA** $112 \times 112$ **Dataset for** $1 \times 1$ **Conv. and UNet Encoders.** This table examines the effect of enabling (✓) or disabling (✗) Adaptive $\sigma_{\mathbf{z}}$ across both $1 \times 1$ Conv. and UNet encoders. Best results for each encoder and metric are highlighted in bold. Results indicate that the proposed adaptive $\sigma_z$ consistently improve results.

| Encoder | Adapt $\sigma_{\mathbf{z}}$ | Radar | | | Temperature | | | Eastward Wind | | | Northward Wind | | |
|---|---|---|---|---|---|---|---|---|---|---|---|---|---|
| | | RMSE ↓ | CRPS ↓ | SSR → 1 | RMSE ↓ | CRPS ↓ | SSR → 1 | RMSE ↓ | CRPS ↓ | SSR → 1 | RMSE ↓ | CRPS ↓ | SSR → 1 |
| $1 \times 1$ Conv. | ✗ | 5.01 | 1.88 | 0.45 | 1.13 | 0.66 | 0.39 | 1.50 | 0.86 | 0.48 | 1.64 | 0.94 | 0.47 |
| | ✓ | **4.82** | **1.66** | **0.72** | **0.99** | **0.59** | **0.47** | **1.42** | **0.78** | **0.68** | **1.58** | **0.89** | **0.66** |
| UNet | ✗ | 5.06 | 1.83 | 0.38 | 1.01 | 0.57 | 0.36 | 1.48 | 0.85 | 0.46 | 1.64 | 0.93 | 0.46 |
| | ✓ | **4.95** | **1.78** | **0.44** | **0.94** | **0.54** | **0.40** | **1.49** | **0.85** | **0.53** | **1.58** | **0.95** | **0.50** |

Table 11: **y conditioning ablation on CWA** $112 \times 112$ **Dataset for** $1 \times 1$ **Conv. and UNet Encoders.** This table assesses the impact of enabling (✓) or disabling (✗) **y** conditioning across both $1 \times 1$ Conv. and UNet encoders. Best results for each encoder and metric are highlighted in bold. Results indicate that for the $1 \times 1$ encoder the conditioning is beneficial while for UNet the opposite stands. This makes sense since the $1 \times 1$ Conv. encoder has only 60 parameters and might not capture all the information in the low-resolution input.

| Encoder | y cond. | Radar | | | Temperature | | | Eastward Wind | | | Northward Wind | | |
|---|---|---|---|---|---|---|---|---|---|---|---|---|---|
| | | RMSE ↓ | CRPS ↓ | SSR → 1 | RMSE ↓ | CRPS ↓ | SSR → 1 | RMSE ↓ | CRPS ↓ | SSR → 1 | RMSE ↓ | CRPS ↓ | SSR → 1 |
| $1 \times 1$ Conv. | ✗ | **4.82** | **1.66** | **0.72** | 0.99 | 0.59 | **0.47** | **1.42** | **0.78** | **0.68** | 1.63 | **0.89** | **0.66** |
| | ✓ | 5.06 | 1.82 | 0.34 | **0.89** | **0.54** | 0.40 | 1.46 | 0.82 | 0.45 | 1.59 | 0.92 | 0.46 |
| UNet | ✗ | 5.06 | 1.83 | 0.38 | 1.01 | 0.57 | 0.36 | 1.48 | **0.85** | 0.46 | 1.64 | 0.93 | 0.46 |
| | ✓ | **4.95** | **1.78** | **0.44** | **0.94** | **0.54** | **0.40** | **1.49** | **0.85** | **0.53** | 1.67 | **0.95** | **0.50** |

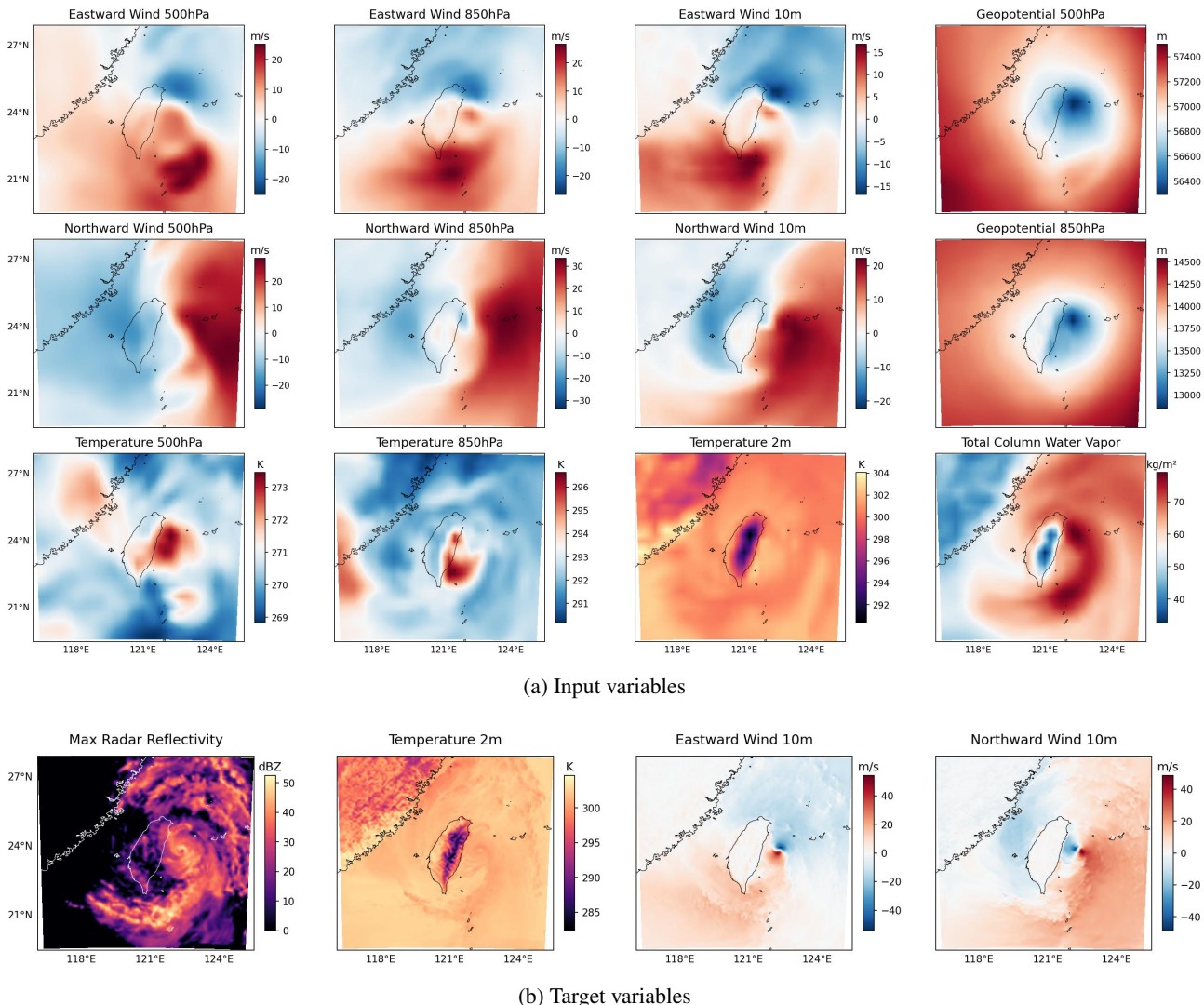

(a) Input variables

(b) Target variables

Figure 7: **Visualization of ERA5-CWA Dataset Variables.** The top row shows input variables such as temperature and wind at coarse-resolution, while the bottom row presents the corresponding fine-resolution target variables. The maximum radar reflectivity is absent from the input variables and must be constructed by the model. This key misalignment between the low- and high-resolution data increases the complexity of the problem beyond standard super-resolution tasks.

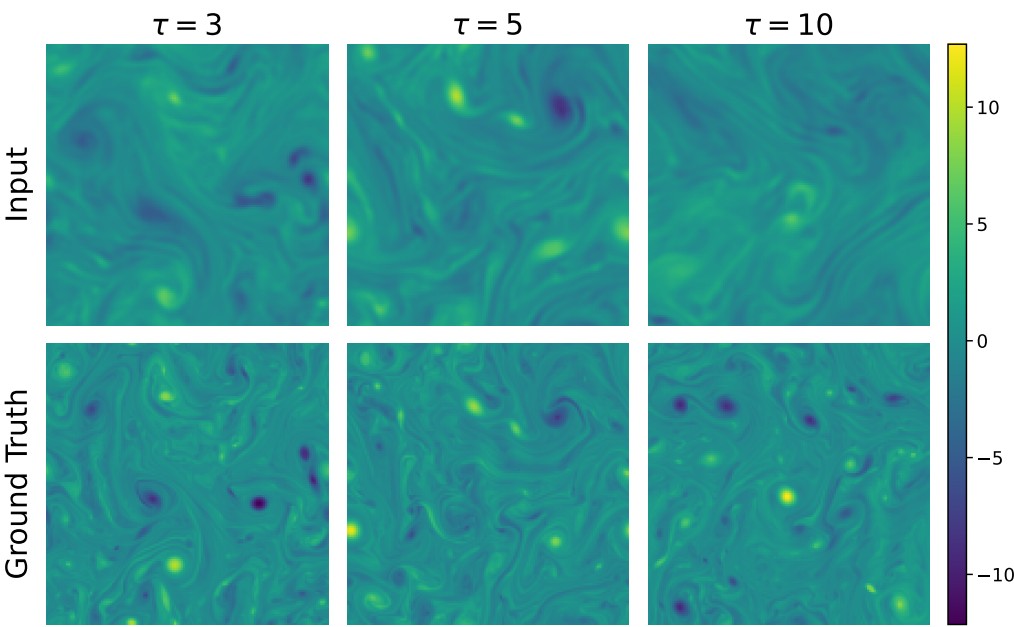

Figure 8: **Kolmogorov Flow Dataset.** Visualization of input and target Kolmogorov Flow dataset for varying levels of misalignment ($\tau = 3, 5, 10$). As $\tau$ increases, the discrepancy between the coarse and fine-resolution fields grows, offering a controlled environment to test downscaling performance.

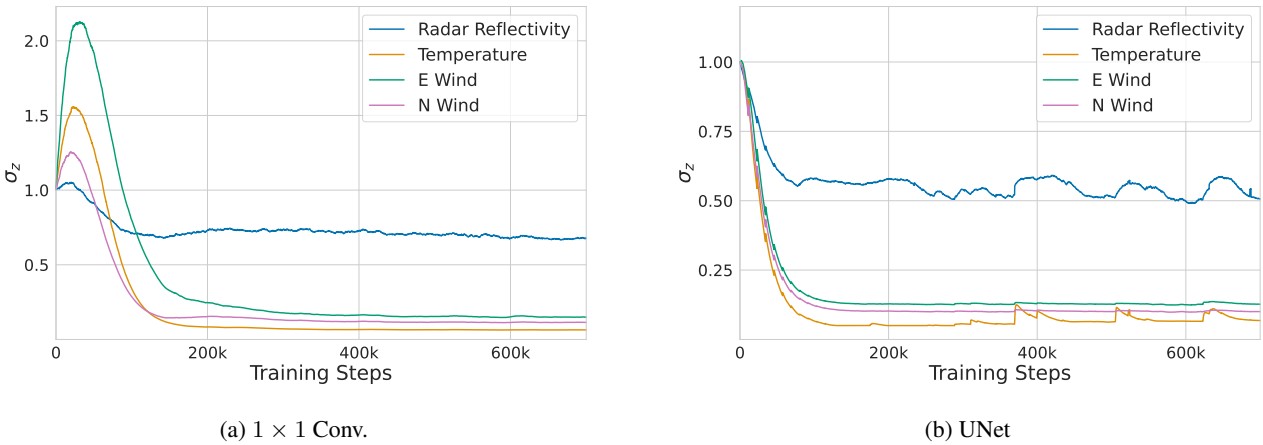

(a) $1 \times 1$ Conv.                 (b) UNet

Figure 9: **Adaptive $\sigma_{\mathbf{z}}$ values over training steps for different channels.** The plot corresponds to the AFM model using $1 \times 1$ Conv. (a) and UNet (b) encoders with $\lambda = 0.25$. With $1 \times 1$ convolution, $\sigma_z$ increases during early training due to high encoder error and subsequently converges as the encoder improves. With UNet, $\sigma_z$ starts decreasing with training. In both cases, radar reflectivity exhibits the highest $\sigma$, indicating its stochastic nature, while temperature shows the lowest, reflecting its deterministic characteristics. The varying sigma values across channels demonstrate why adaptive noise scaling is effective in managing misaligned data with differing levels of stochasticity.

Table 12: **Complete ablation results for the CWA** $112 \times 112$ **dataset.** Best two models in each variable/metric in bold.

| Model | $\lambda$ | Adapt. $\sigma_{max}$ | Use $\mathbf{x}_{low}$ | Radar RMSE ↓ | Radar CRPS ↓ | Radar SSR → 1 | Temp. RMSE | Temp. CRPS | Temp. SSR | East. RMSE | East. CRPS | East. SSR | North. RMSE | North. CRPS | North. SSR |
|---|---|---|---|---|---|---|---|---|---|---|---|---|---|---|---|
| AFM $1 \times 1$ **Conv.** | 0.00 | ✗ | ✗ | 5.16 | 1.88 | 0.44 | 1.12 | 0.65 | 0.36 | 1.53 | 0.88 | 0.47 | 1.72 | 0.98 | 0.46 |
| | | ✓ | ✗ | **4.82** | **1.66** | **0.72** | 0.99 | 0.59 | **0.47** | **1.42** | **0.78** | **0.68** | 1.63 | **0.89** | **0.66** |
| | | | ✓ | 5.06 | 1.82 | 0.34 | 0.87 | 0.52 | **0.44** | 1.44 | 0.81 | 0.49 | 1.60 | **0.89** | 0.49 |
| | 0.25 | ✗ | ✗ | 5.10 | 1.83 | 0.42 | **0.85** | **0.50** | 0.43 | 1.46 | 0.84 | 0.46 | 1.63 | 0.92 | 0.46 |
| | | ✓ | ✗ | 5.11 | 1.85 | 0.37 | 0.89 | 0.56 | 0.28 | 1.53 | 0.92 | 0.36 | 1.71 | 1.02 | 0.34 |
| | | | ✓ | 5.12 | 1.87 | 0.29 | 0.89 | 0.54 | 0.40 | 1.46 | 0.84 | 0.45 | 1.59 | 0.90 | 0.46 |
| | 1.00 | ✗ | ✗ | 5.04 | 1.83 | 0.40 | 1.05 | 0.60 | 0.34 | 1.50 | 0.88 | 0.44 | 1.67 | 0.95 | 0.44 |
| | | ✓ | ✗ | 5.11 | 1.85 | 0.36 | 0.92 | 0.58 | 0.30 | 1.52 | 0.92 | 0.35 | 1.71 | 1.01 | 0.34 |
| | | | ✓ | 5.12 | 1.88 | 0.30 | 0.86 | **0.50** | 0.41 | **1.43** | 0.82 | 0.46 | **1.58** | 0.90 | 0.45 |
| AFM **UNet** | 0.00 | ✗ | ✗ | 5.06 | 1.83 | 0.38 | 1.01 | 0.57 | 0.36 | 1.48 | 0.85 | 0.46 | 1.64 | 0.93 | 0.46 |
| | | ✓ | ✗ | 5.13 | 1.84 | 0.43 | **0.85** | 0.52 | 0.37 | **1.43** | **0.81** | 0.50 | 1.60 | 0.89 | 0.48 |
| | | | ✓ | 5.07 | 1.84 | 0.36 | **0.85** | 0.52 | 0.34 | 1.44 | 0.83 | 0.43 | 1.60 | 0.91 | 0.43 |
| | 0.25 | ✗ | ✗ | 5.01 | 1.88 | 0.45 | 1.13 | 0.66 | 0.39 | 1.50 | 0.86 | 0.48 | 1.64 | 0.94 | 0.47 |
| | | ✓ | ✗ | 5.01 | 1.82 | 0.32 | **0.85** | 0.52 | 0.31 | **1.43** | 0.85 | 0.40 | **1.58** | 0.93 | 0.39 |
| | | | ✓ | **4.95** | **1.78** | **0.44** | 0.94 | 0.54 | 0.40 | 1.49 | 0.85 | **0.53** | 1.67 | 0.95 | **0.50** |
| | 1.00 | ✗ | ✗ | 5.11 | 1.89 | 0.41 | 1.07 | 0.62 | 0.36 | 1.53 | 0.91 | 0.41 | 1.74 | 1.03 | 0.40 |
| | | ✓ | ✗ | 5.04 | 1.85 | 0.29 | **0.84** | 0.53 | 0.26 | 1.45 | 0.87 | 0.35 | 1.62 | 0.97 | 0.34 |
| | | | ✓ | 5.07 | 1.85 | 0.31 | 0.88 | 0.54 | 0.29 | 1.46 | 0.88 | 0.38 | 1.65 | 0.98 | 0.37 |

Table 13: **Performance Comparison of Models on CWA** $112 \times 112$ **Dataset.** The AFM model has a $1 \times 1$ Conv. encoder, $\lambda = 0$, adaptive $\sigma_z$ and no $y$ conditioning. Overall, the AFM model exhibits strong performance across different metrics and variables, particularly excelling in its calibration (variability). Best results for each metric are highlighted in bold. Note that for deterministic models, CRPS equals MAE.

| | Model | CFM | CDM | UNet | AFM |
|---|---|---|---|---|---|
| **Radar** | RMSE↓ | 5.06 | 4.95 | 4.94 | **4.82** |
| | CRPS↓ | 1.84 | 1.74 | - | **1.66** |
| | MAE↓ | **2.41** | 2.49 | 2.45 | 2.63 |
| | SSR → 1 | 0.36 | 0.52 | - | **0.72** |
| **Temperature** | RMSE | 0.86 | 0.87 | 0.87 | 0.99 |
| | CRPS | **0.50** | 0.52 | - | 0.59 |
| | MAE | 0.64 | 0.64 | 0.64 | 0.74 |
| | SSR | 0.45 | 0.38 | - | **0.47** |
| **East. Wind** | RMSE | **1.42** | 1.44 | **1.42** | **1.42** |
| | CRPS | 0.81 | 0.81 | - | **0.78** |
| | MAE | **1.04** | 1.05 | 1.05 | 1.05 |
| | SSR | 0.48 | 0.49 | - | **0.68** |
| **North. Wind** | RMSE | **1.59** | **1.59** | 1.60 | 1.63 |
| | CRPS | **0.89** | **0.89** | - | **0.89** |
| | MAE | **1.14** | **1.14** | 1.16 | 1.19 |
| | SSR | 0.47 | 0.48 | - | **0.66** |

Table 14: **AFM ablations for $\lambda$ on the CWA weather downscaling task at full resolution** $448 \times 448$. For this ablation, a $1 \times 1$ Conv. encoder was used, $\sigma_z$ was set to 1, and no **y** conditioning was employed. Overall, $\lambda = 0$ seems to produce better estimates except for temperature, whose deterministic nature benefits from the added regularization.

| Variable | Metric | $\lambda$ | | | | | | |
|---|---|---|---|---|---|---|---|---|
| | | **0.0** | **0.01** | **0.1** | **0.25** | **0.5** | **1.0** | **2.5** |
| Radar | RMSE ↓ | **4.90** | 5.11 | 5.03 | 4.97 | 5.00 | 5.25 | 5.27 |
| | CRPS ↓ | **1.78** | 1.92 | 1.88 | 1.85 | 1.87 | 1.97 | 1.96 |
| | MAE ↓ | **2.42** | 2.66 | 2.47 | 2.47 | 2.47 | 2.59 | 2.75 |
| | SSR→1 | 0.44 | 0.49 | 0.39 | 0.42 | 0.41 | 0.39 | **0.52** |
| Temperature | RMSE ↓ | 1.00 | 1.00 | 0.87 | 0.86 | **0.82** | 1.00 | 0.85 |
| | CRPS ↓ | 0.52 | 0.62 | 0.54 | 0.52 | **0.48** | 0.56 | **0.48** |
| | MAE ↓ | 0.67 | 0.78 | 0.66 | 0.64 | **0.60** | 0.68 | 0.62 |
| | SSR → 1 | 0.47 | **0.48** | 0.38 | 0.40 | 0.41 | 0.32 | 0.45 |
| East. Wind | RMSE ↓ | **1.44** | 1.52 | 1.49 | 1.48 | 1.49 | 1.55 | 1.48 |
| | CRPS ↓ | **0.80** | 0.86 | 0.87 | 0.87 | 0.88 | 0.92 | 0.86 |
| | MAE ↓ | **1.07** | 1.13 | 1.09 | 1.09 | 1.09 | 1.14 | 1.08 |
| | SSR → 1 | **0.61** | 0.55 | 0.42 | 0.40 | 0.41 | 0.38 | 0.43 |
| North. Wind | RMSE ↓ | **1.61** | **1.61** | 1.66 | 1.67 | 1.67 | 1.72 | 1.65 |
| | CRPS ↓ | **0.88** | **0.88** | 0.95 | 0.96 | 0.96 | 1.00 | 0.95 |
| | MAE ↓ | **1.17** | **1.17** | 1.19 | 1.20 | 1.19 | 1.24 | 1.18 |
| | SSR → 1 | 0.58 | **0.61** | 0.41 | 0.41 | 0.40 | 0.38 | 0.43 |

Table 15: **AFM ablations for encoder size on the CWA weather downscaling task at full resolution** $448 \times 448$. For this ablation, we use $\lambda = 0$, $\sigma_z = 1$, and no **y** conditioning. Larger encoders improve performance for complex spatial data like radar, while for temperature and wind data, smaller encoders are adequate and sometimes even slightly better. The optimal encoder size depends on the specific variable being predicted but overall the difference are not significant.

| Variable | Metric | Encoder | | | |
|---|---|---|---|---|---|
| | | **L** | **M** | **S** | **XS** |
| Radar | RMSE ↓ | **4.93** | 4.96 | 4.98 | 4.98 |
| | CRPS ↓ | **1.82** | 1.84 | 1.85 | 1.85 |
| | MAE ↓ | **2.44** | 2.52 | 2.52 | 2.46 |
| | SSR→1 | 0.40 | **0.43** | 0.42 | 0.39 |
| Temperature | RMSE ↓ | 1.01 | 1.00 | **0.99** | 1.02 |
| | CRPS ↓ | 0.55 | **0.54** | **0.54** | 0.56 |
| | MAE ↓ | **0.68** | **0.68** | **0.68** | 0.70 |
| | SSR → 1 | 0.34 | 0.38 | **0.40** | 0.38 |
| East. Wind | RMSE ↓ | 1.48 | 1.48 | **1.47** | 1.50 |
| | CRPS ↓ | 0.85 | **0.83** | **0.83** | 0.86 |
| | MAE ↓ | 1.10 | **1.08** | **1.08** | 1.11 |
| | SSR → 1 | 0.49 | **0.53** | **0.53** | 0.51 |
| North. Wind | RMSE ↓ | 1.64 | 1.64 | **1.63** | 1.66 |
| | CRPS ↓ | 0.92 | 0.92 | **0.91** | 0.93 |
| | MAE ↓ | 1.19 | 1.20 | **1.19** | 1.21 |
| | SSR → 1 | 0.48 | **0.51** | 0.52 | 0.50 |

Table 16: **AFM ablations for $y$ conditioning for the CWA weather downscaling task at full resolution** $448 \times 448$. For this ablation, we use adaptive $\sigma_z$, $\lambda = 0.25$ and a UNet encoder guided by the ablations in the $112 \times 112$ resolution. Including $\mathbf{y}$ conditioning consistently improves performance across all metrics. RMSE, CRPS, and MAE are lower when $\mathbf{y}$ conditioning is used, and SSR values are closer to 1.

| Variable | Metric | y conditioning | |
|---|---|---|---|
| | | ✗ | ✓ |
| Radar | **RMSE** $\downarrow$ | 5.09 | **4.90** |
| | **CRPS** $\downarrow$ | 1.88 | **1.80** |
| | **MAE** $\downarrow$ | 2.33 | **2.24** |
| | **SSR** $\rightarrow 1$ | 0.24 | **0.25** |
| Temperature | **RMSE** $\downarrow$ | 0.92 | **0.89** |
| | **CRPS** $\downarrow$ | 0.55 | **0.50** |
| | **MAE** $\downarrow$ | 0.67 | **0.64** |
| | **SSR** $\rightarrow 1$ | 0.33 | **0.43** |
| East. Wind | **RMSE** $\downarrow$ | 1.49 | **1.45** |
| | **CRPS** $\downarrow$ | 0.91 | **0.86** |
| | **MAE** $\downarrow$ | 1.10 | **1.07** |
| | **SSR** $\rightarrow 1$ | 0.34 | **0.41** |
| North. Wind | **RMSE** $\downarrow$ | 1.66 | **1.61** |
| | **CRPS** $\downarrow$ | 1.00 | **0.94** |
| | **MAE** $\downarrow$ | 1.20 | **1.18** |
| | **SSR** $\rightarrow 1$ | 0.33 | **0.41** |

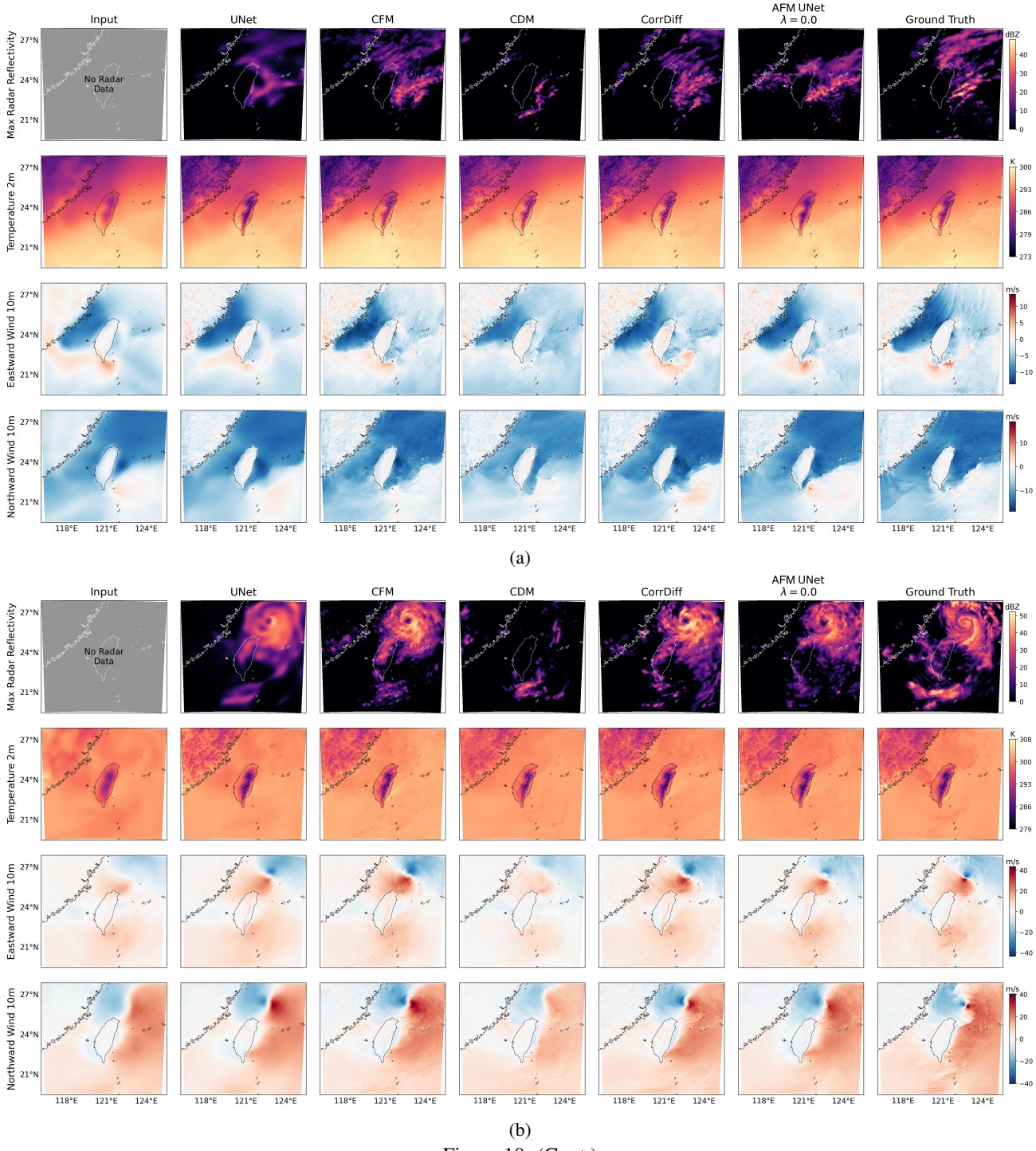

(a)

(b)

Figure 10: (Cont.)

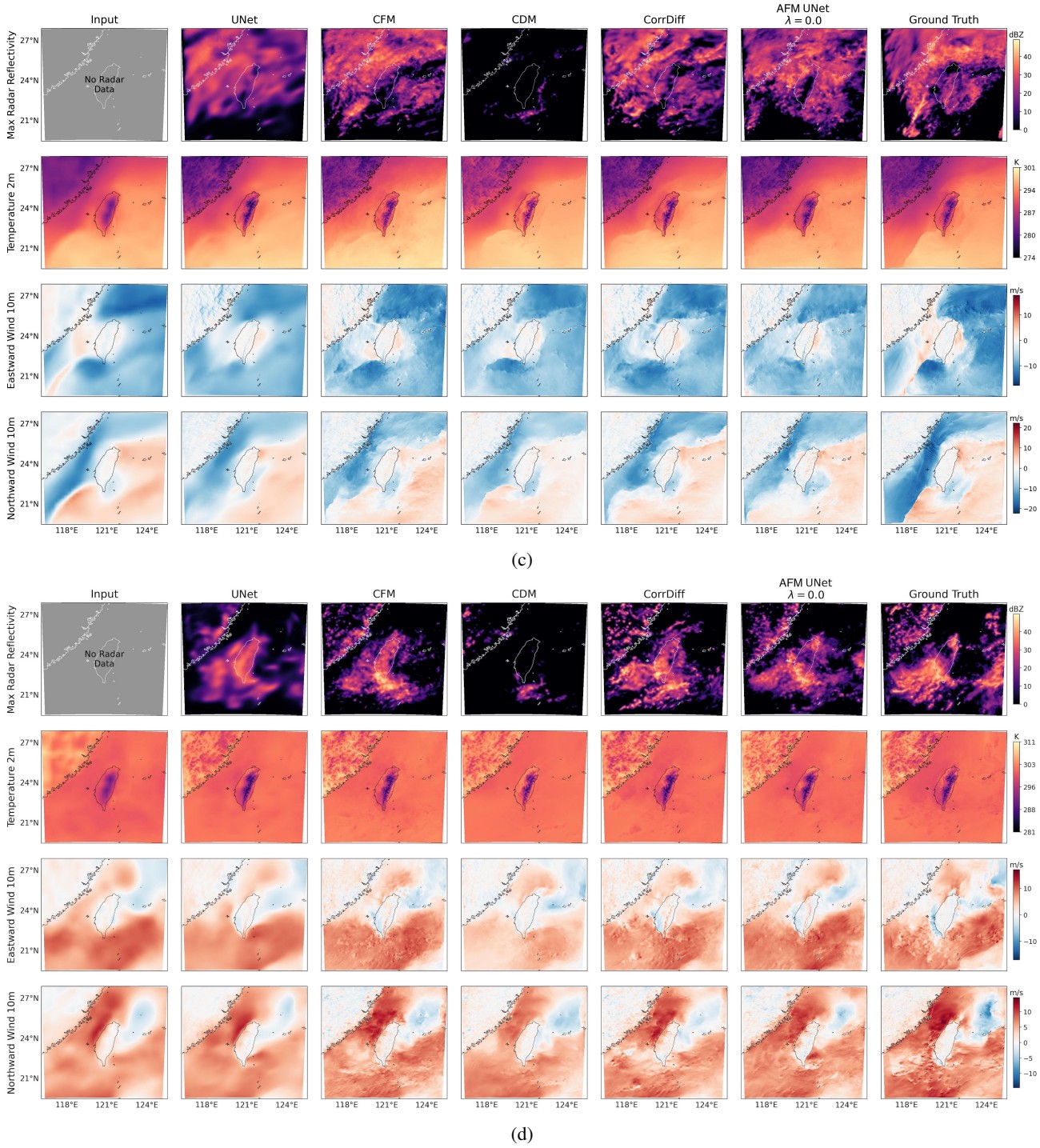

(c)

(d)

Figure 10: (Cont.)

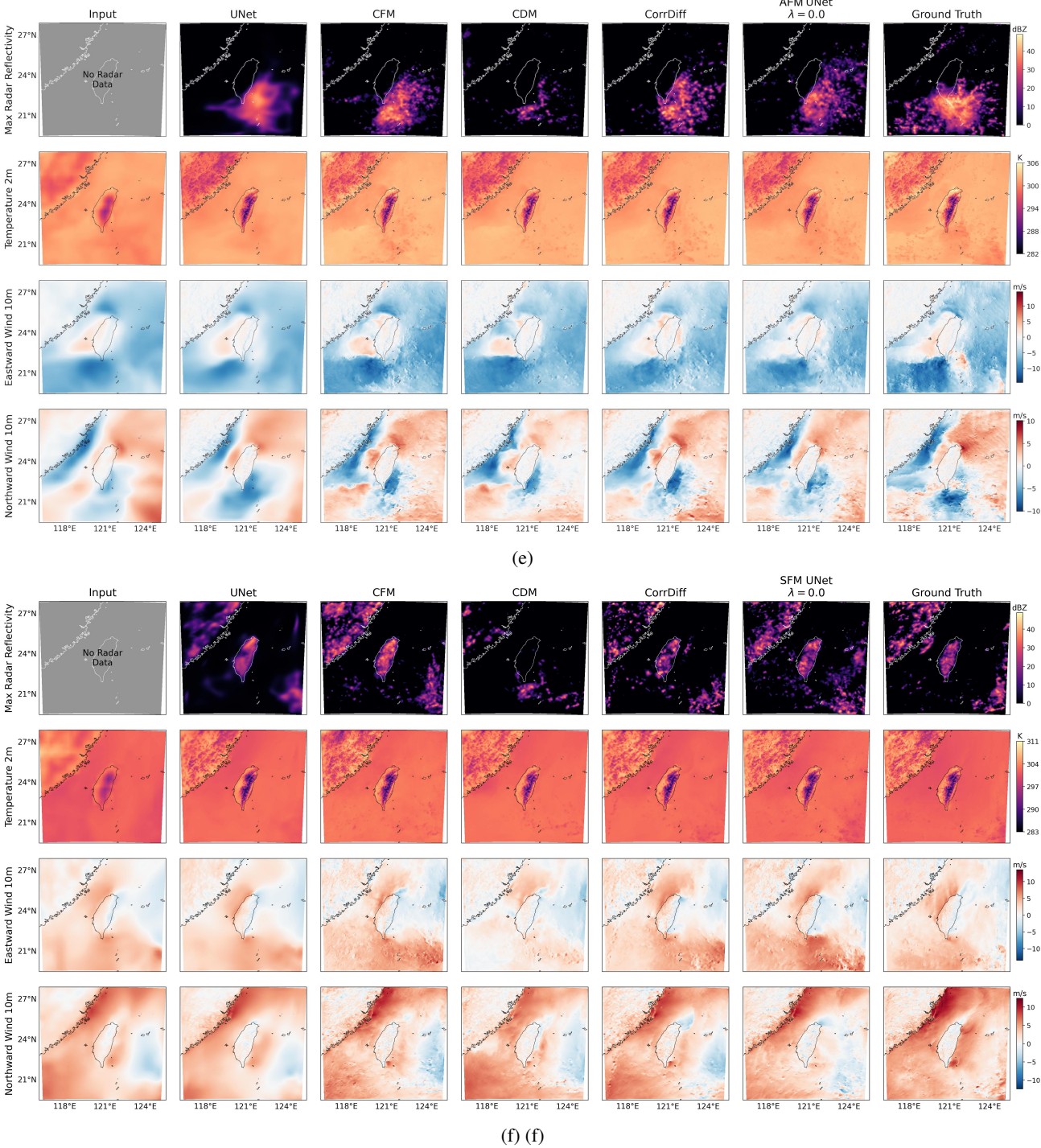

Figure 10: **(a-e) Visual Comparison of All Models for CWA Weather Data.** The AFM model demonstrates superior reconstruction quality, particularly in capturing fine-scale details, while other baselines show blurring or misalignment in key areas. (a-e) show the results for different models.

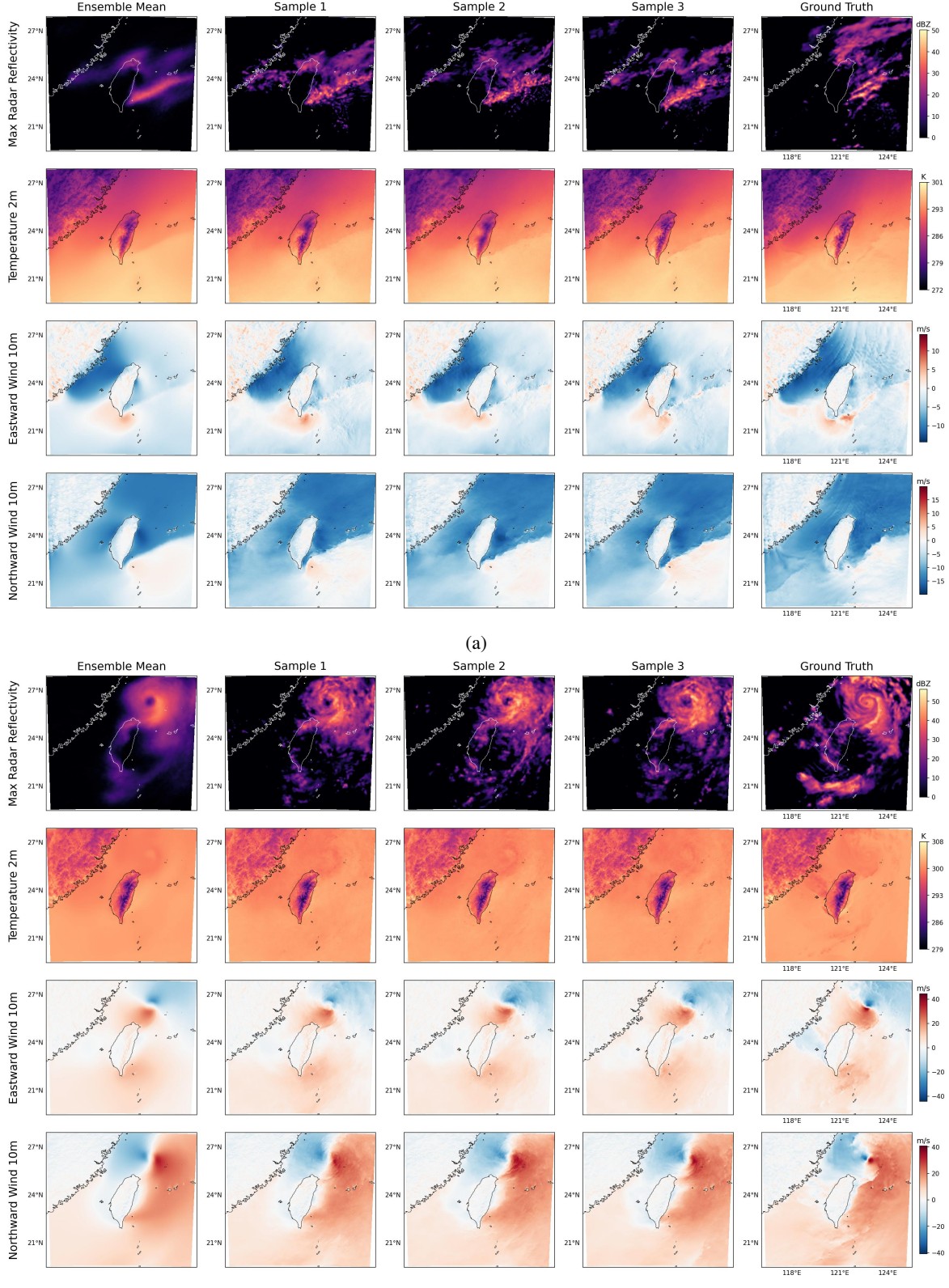

(a)

(b)
Figure 11: (Cont.)

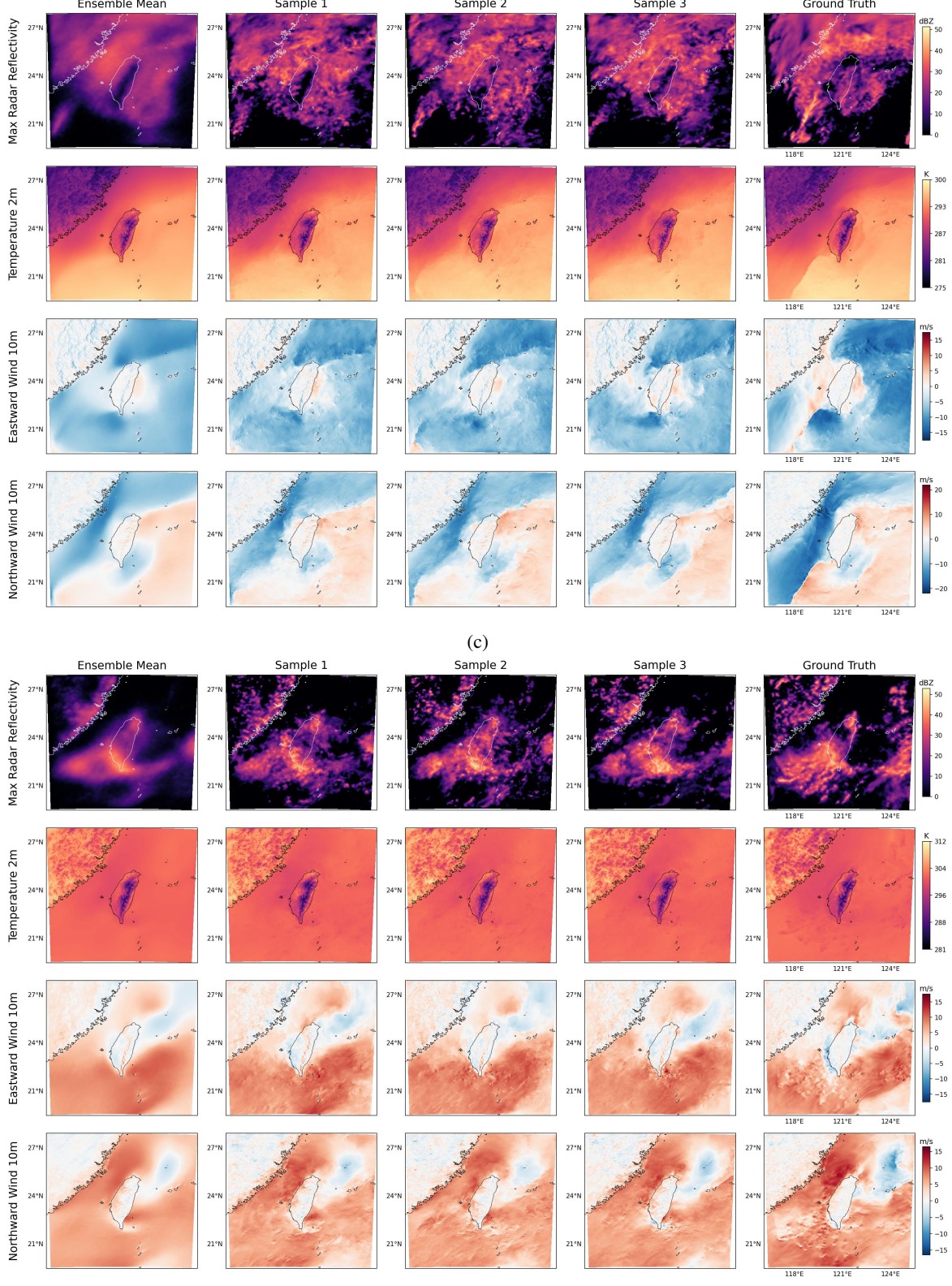

(c)

(d)

Figure 11: (Cont.)

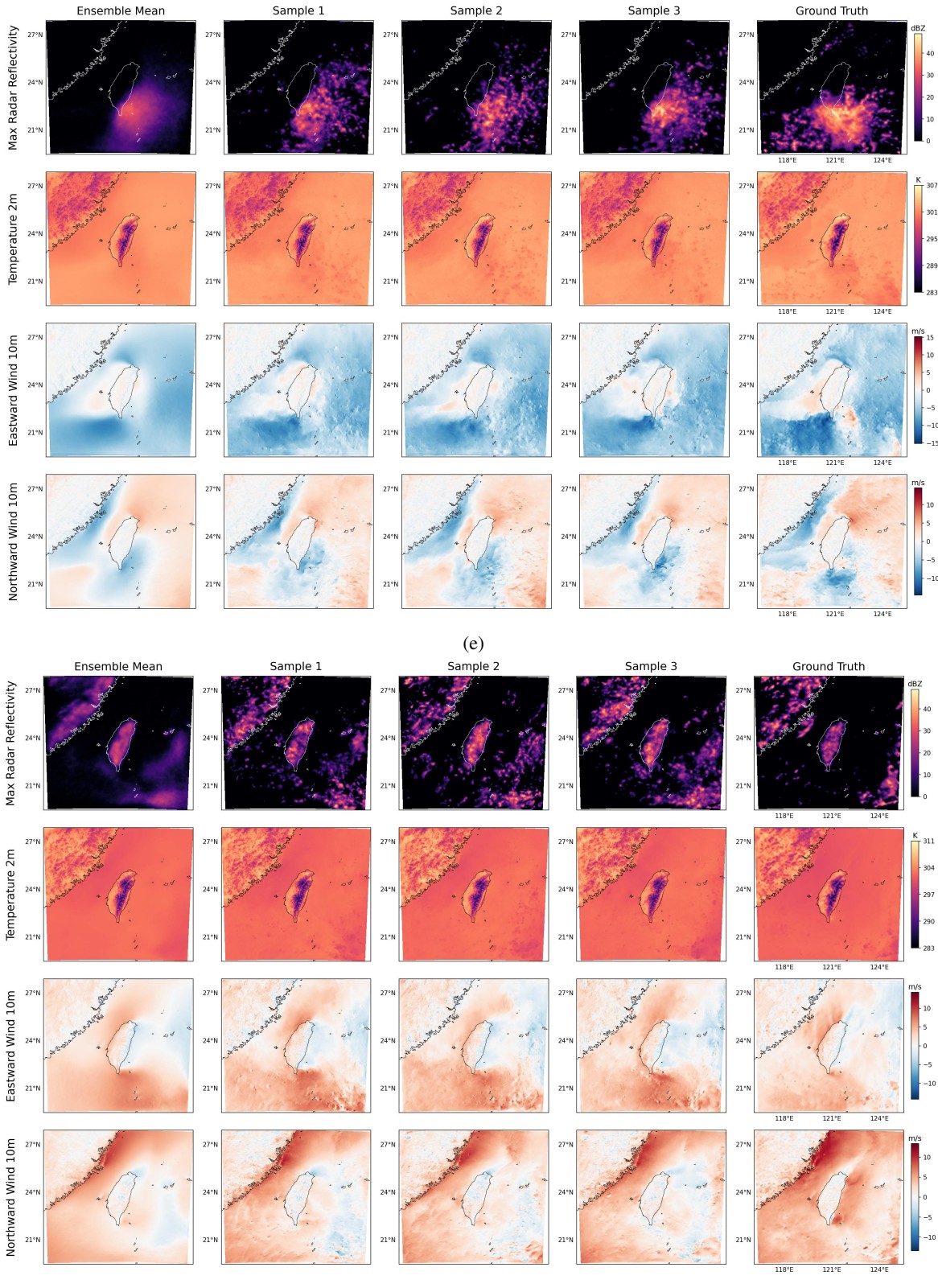

(e)

(f)

Figure 11: **Ensemble predictions for CWA Weather Data.** Results demonstrate AFM's ability to capture variable dynamics. (a-e) show the results for different points in time.

## K. Kolmogorov Flow Ablations

Different hyperparameters of both the dataset and the model are ablated, with the resulting metrics reported in Table 17. The results show that, in this scenario, where the data is relatively misaligned, a smaller encoder tends to achieve better generalization performance, possibly due to its capacity to focus on the most relevant features while avoiding overfitting. Additionally, the experiments demonstrate that conditioning the AFM with the coarse-resolution input enhances the predictive skill, highlighting the importance of incorporating multiscale information for improved downscaling accuracy. These findings provide valuable insights into the optimal model configurations for handling misaligned data.

Table 17: **Kolmogorov Flow Ablation Study for AFM:** This table examines the effect of different hyperparameters on performance across misalignment levels ($\tau$). A smaller encoder with conditioning consistently performs better for highly misaligned data. Additionally, adaptive noise scaling ($\sigma_z$) enhances performance when conditioning AFM on coarse-resolution input data (**y**).

| | Encoder | $1 \times 1$conv | | | | UNet | | | |
|---|---|---|---|---|---|---|---|---|---|
| | Adapt. $\sigma_z$ | ✗ | | ✓ | | ✗ | | ✓ | |
| $\tau$ | y cond. | ✗ | ✓ | ✗ | ✓ | ✗ | ✓ | ✗ | ✓ |
| | RMSE ↓ | 1.22 | 0.91 | 1.22 | **0.73** | 1.15 | 1.11 | 1.17 | 1.17 |
| 3 | CRPS ↓ | 0.63 | 0.48 | 0.62 | **0.37** | 0.65 | 0.63 | 0.69 | 0.70 |
| | MAE ↓ | 0.83 | 0.65 | 0.83 | **0.51** | 0.81 | 0.78 | 0.83 | 0.84 |
| | SSR → 1 | 0.56 | 0.58 | 0.58 | **0.62** | 0.37 | 0.34 | 0.31 | 0.30 |
| | RMSE ↓ | 1.17 | 0.78 | 1.17 | **0.76** | 1.16 | 1.01 | 1.09 | 1.07 |
| 5 | CRPS ↓ | 0.62 | 0.42 | 0.61 | **0.40** | 0.67 | 0.58 | 0.66 | 0.64 |
| | MAE ↓ | 0.82 | 0.56 | 0.82 | **0.54** | 0.83 | 0.72 | 0.80 | 0.78 |
| | SSR → 1 | 0.58 | 0.58 | **0.60** | 0.58 | 0.35 | 0.36 | 0.33 | 0.33 |
| | RMSE ↓ | 1.36 | **1.06** | 1.39 | 1.09 | 1.35 | 1.36 | 1.36 | 1.28 |
| 10 | CRPS ↓ | 0.71 | **0.57** | 0.78 | 0.65 | 0.79 | 0.79 | 0.80 | 0.74 |
| | MAE ↓ | 0.96 | 0.78 | 0.98 | **0.77** | 0.98 | 0.99 | 1.00 | 0.95 |
| | SSR → 1 | **0.69** | 0.63 | 0.43 | 0.23 | 0.36 | 0.37 | 0.38 | 0.45 |

## L. Kolmogorov Flow Ensemble Analysis

Representative KF samples along with the generated ensemble members are depicted in Figs. 14a, 14b, and 14c. These figures illustrate the variability captured by the ensemble across different forecast lead times. The diversity in the ensemble members indicates the model's ability to represent the inherent uncertainty in the system. Furthermore, the alignment of the ensemble mean with the observed samples suggests that the model not only captures the central tendency but also effectively characterizes the stochastic nature of the dynamics.

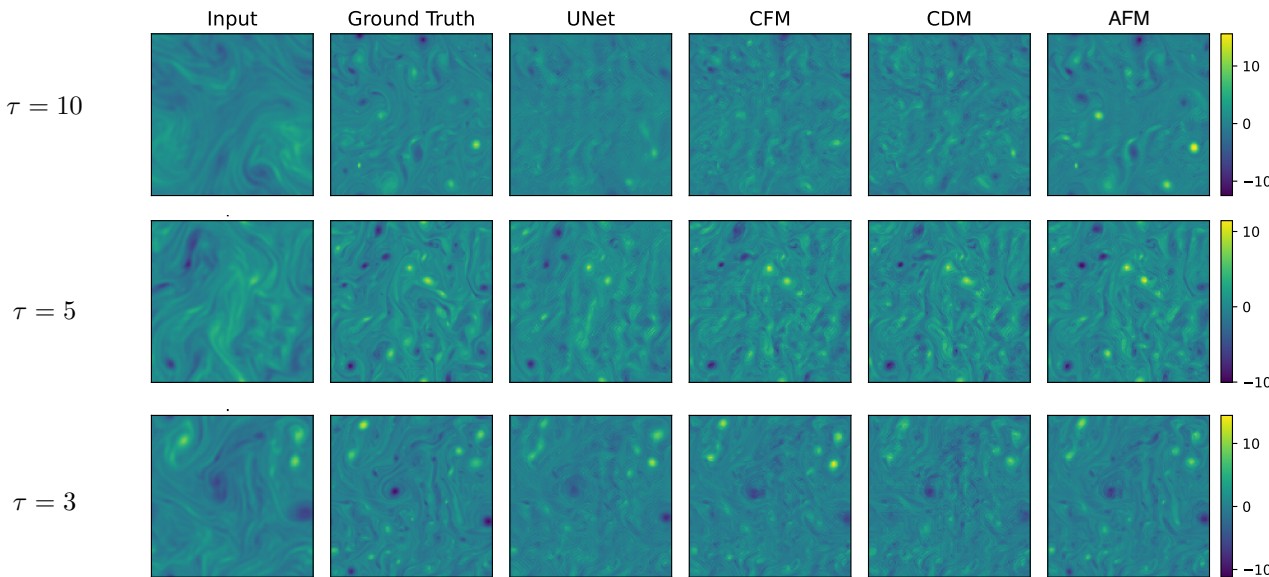

Figure 12: **AFM vs. Baselines for Different Misalignment Levels in Kolmogorov Flow Downscaling:** Each row corresponds to a different misalignment level $\tau$ (left side). From top to bottom, the rows represent $\tau = 10$, $\tau = 5$, and $\tau = 3$. As misalignment increases, the AFM significantly outperforms baseline models by generating samples that better align with the target distribution. Additionally, note the presence of high-frequency artifacts in the baseline models, which are more noticeable when the figures are zoomed in.

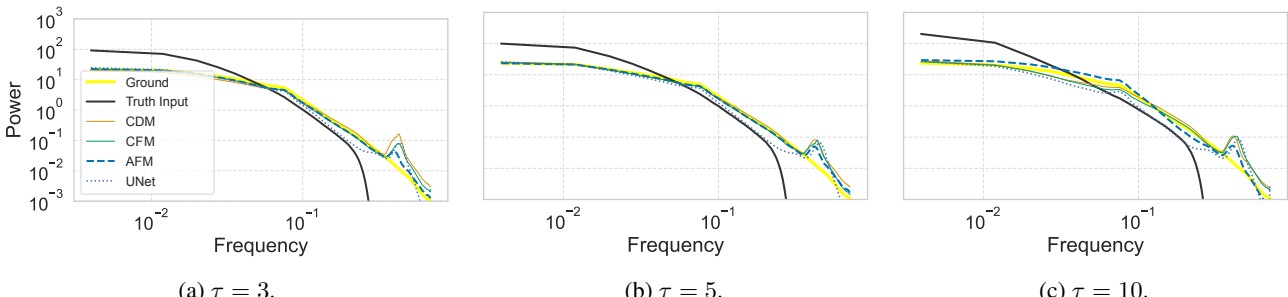

(a) $\tau = 3$.  (b) $\tau = 5$.  (c) $\tau = 10$.

Figure 13: **AFM spectra vs. baselines for the Kolmogorov Flow.** AFM maintains superior fidelity to the ground truth across different $\tau$ values, highlighting its robustness in preserving both small and large-scale structures under various misalignment conditions. The small bump around the middle is caused by the energy that is added in the system (see Appendix G.2).

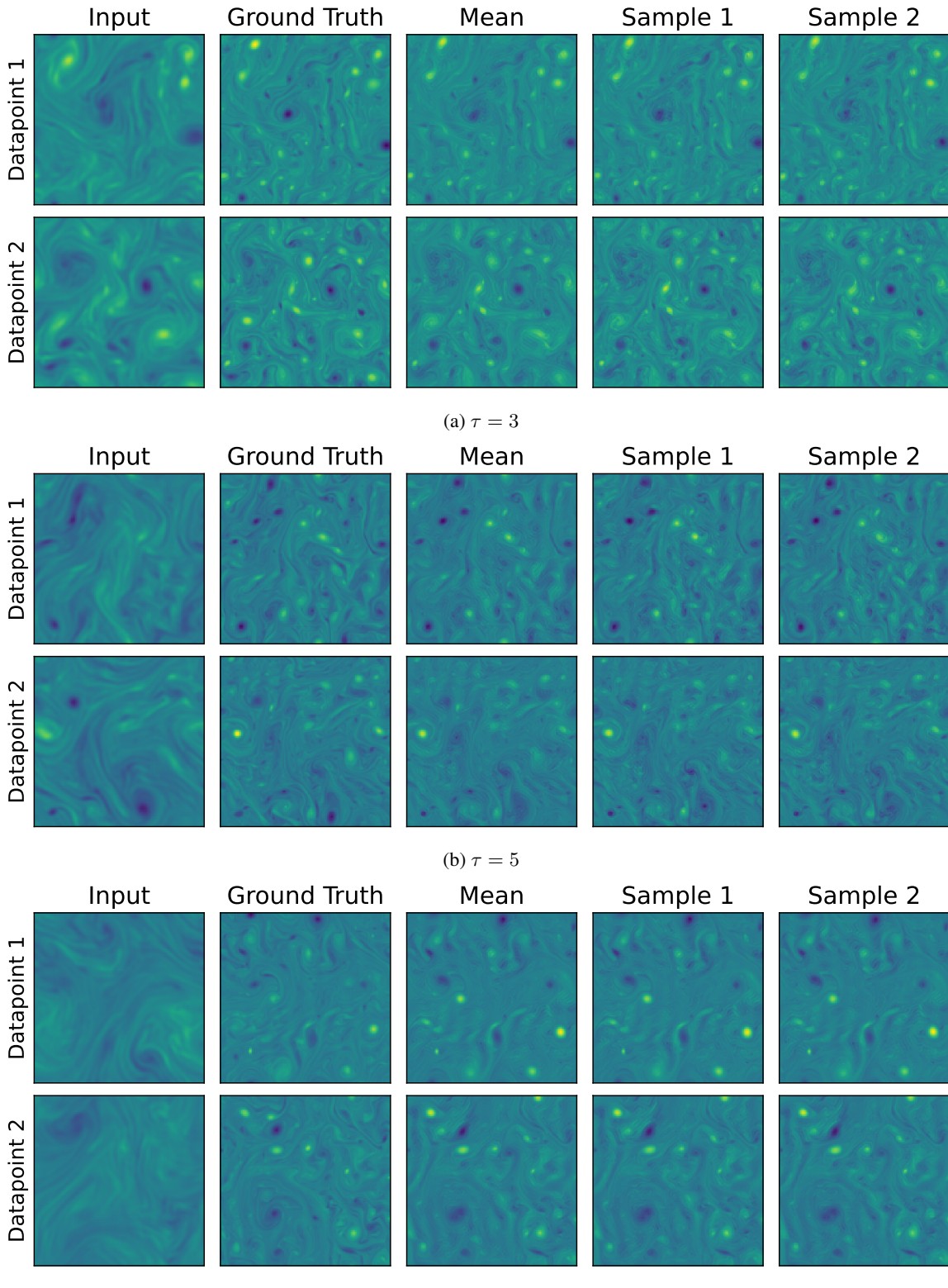

(a) $\tau = 3$

(b) $\tau = 5$

(c) $\tau = 10$

Figure 14: **Ensemble Predictions of AFM for Kolmogorov Flow at Different $\tau$ Values.** AFM ensemble predictions are shown for different $\tau$ values ($\tau = 3, 5, 10$), illustrating the model's ability to capture the variability and dynamics of the Kolmogorov flow across increasing levels of misalignment.

