# OpenReview forum: "Adaptive Flow Matching for Resolving Small-Scale Physics"
_ICML.cc/2025/Conference — ICML 2025 poster_

### Official Review · Reviewer_X7XZ · 2025-03-12

**Overall Recommendation:** 3

**Summary:**

Applying conditional diffusion (CDM) and flow matching (FM) to natural images is very effective for super-resolving small-scale details, like the image semantic- or geometric information. However, CDM and FM will have difficulties with physical sciences, particularly for weather, mainly due to 1) spatially misaligned input-output (super-res weather input); 2) misaligned and distinct input-output channels (various channels sensor data); 3) stochasticity in channels resulting in multi-scales issue. To alleviate these challenges for weather application scenarios, this paper proposes first to encode various inputs to a latent distribution that is closer to the target and reduce input differences. Then the authors map the small-scale physics with Flow Matching. So Flow Matching adds stochastic details while the encoder adds deterministic ones, an adaptive noise scaling mechanism is utilized to dynamically adjust the noise scale at specific iterations to reduce the residual error and maintain the generalization. Extensive experiments on benchmarks, like real-world weather data  (25 km to 2 km scales in Taiwan) and in synthetic Kolmogorov flow datasets, validate the effectiveness of the proposed method, Adaptive Flow Matching.

**Claims And Evidence:**

Yes, the authors have supported necessary claims and evidence with step-by-step proofs.

**Essential References Not Discussed:**

No.

**Experimental Designs Or Analyses:**

Regarding Fig 2, I guess the SFM shall be the method proposed in this paper.

When we compare the visually generated images between SFM and CorrDiff, it seems CorrDiff is better. Can the authors provide more convincing explanations about the advantages of CorrDiff?

I am also curious whether the proposed adaptive flow matching works for nature images, and I do expect to see such validated experiments to make this paper stronger.

**Methods And Evaluation Criteria:**

Although the main contribution of this paper is to apply conditional diffusion and flow matching into a very downstream task, whether physical science, they propose some technical methods to resolve the problems raised by the application tasks and obtain competitive performances.

**Other Comments Or Suggestions:**

1. SFM in Figures shall be well denoted, since we may think it shall be AFM.

2. How do the main contributions of adaptive flow matching from interpolating the latent from common flow-matching based nature image methods?

**Other Strengths And Weaknesses:**

Strengths:

1. The presentation of this paper is well and extensive experiments on weather scenario validation have been conducted.

2. The writing of this paper is also well-following.

Weaknesses:

1. The applications to weather scenarios are limited, or we need to see large-scale validations to make this paper more convincing.

2. Whether the proposed adaptive flow matching works for the nature image task and needs to conduct further experiments.

**Questions For Authors:**

Please see my overall comments and questions above, and I suggest the authors address them well.

**Relation To Broader Scientific Literature:**

This paper aims to apply CDM and FM to physical science weather and conduct experiments on regional weather data, from 20km to 2km, and synthesis data. Though they obtain competitive performances and support theoretical proofs to demonstrate their method, it is not clear whether this method can be generalized or employed to other tasks, both physical science and nature image tasks. So this leaves a doubt about the wide application of this paper and I suggest the author can make some more demonstrations.

**Theoretical Claims:**

I have not checked each proof detail, presented in the Appendix, but the proofs in the main paper sound reasonable. For Eq (9), imposing interpolation like this formulation has been widely used in flow-matching based diffusion methods for nature image tasks.

---

> ### Author Rebuttal · Authors · 2025-04-01
>
> We thank the reviewer for their remarks and for recognizing the extensive experimental validation of our work.
>
> *1. When we compare the visually generated images between SFM and CorrDiff, it seems CorrDiff is better. Can the authors provide more convincing explanations about the advantages of CorrDiff?*
>
>
> **Response.** While a single visual sample (e.g., Fig. 2) may suggest CorrDiff looks better, such examples can be misleading and not representative. A more robust comparison is provided in Fig. 3, where the spectral analysis shows that CorrDiff underperforms AFM at both low and high frequencies. Moreover, quantitative results in Table 2 demonstrate that AFM achieves better reconstruction accuracy and significantly improved uncertainty calibration.  In the revised manuscript appendix, we have included another case in each of Figures 9 and 10 to shed more light in this comparison.
>
> ---
>
> *2a. Though they obtain competitive performances and support theoretical proofs to demonstrate their method, it is not clear whether this method can be generalized or employed to other tasks, both physical science and nature image tasks. So this leaves a doubt about the wide application of this paper and I suggest the author can make some more demonstrations.*
>
> *2b.I am also curious whether the proposed adaptive flow matching works for nature images, and I do expect to see such validated experiments to make this paper stronger.*
>
> *2c.The applications to weather scenarios are limited, or we need to see large-scale validations to make this paper more convincing.*
>
> *2d. How do the main contributions of adaptive flow matching from interpolating the latent from common flow-matching based nature image methods?*
>
> **Response.** Thank you for the suggestion. While our method was motivated by challenges in scientific data, it makes no domain-specific assumptions and is broadly applicable. To demonstrate generality, we include two diverse testbeds: (1) real-world weather downscaling using Taiwan’s CWA dataset, and (2) a synthetic PDE-based Kolmogorov flow dataset. Extending AFM to natural image tasks is a promising direction, but beyond the scope of this already extensive study. We have added this discussion to the conclusion of the revised manuscript.
>
> ---
>
> *4a. Regarding Fig 2, I guess the SFM shall be the method proposed in this paper.*
>
> *4b. SFM in Figures shall be well denoted, since we may think it shall be AFM.*
>
> **Response.** Thank you for noting this. We have fixed the AFM label in the figures of the revised manuscript.
>
> We hope our responses and revisions have adequately addressed your remarks.

---

> > ### Comment · Reviewer_X7XZ · 2025-04-04
> >
> > Somehow, the authors have addressed most of my concerns, hence I keep my original rating for this paper.

---

### Official Review · Reviewer_zyDu · 2025-03-13

**Overall Recommendation:** 3

**Summary:**

The paper introduces stochastic flow matching (SFM) for super-resolving small-scale physics in weather data, tackling challenges such as data misalignment, multiscale dynamics, and limited data availability. The approach employs an encoder to project coarse-resolution inputs into a latent space, followed by flow matching to generate fine-resolution outputs. Additionally, adaptive noise scaling is used to balance deterministic and stochastic components. Experimental results show that SFM surpasses existing methods, including conditional diffusion and flow models.

**Claims And Evidence:**

Yes.

**Essential References Not Discussed:**

No.

**Experimental Designs Or Analyses:**

I reviewed the experimental design and analyses presented in the paper, including the methodology for super-resolving small-scale physics in weather data using stochastic flow matching (SFM). Specifically, I examined how the authors addressed challenges such as data misalignment, multiscale dynamics, and limited data availability. I also assessed the effectiveness of the encoder-latent space mapping, flow matching process, and adaptive noise scaling in balancing deterministic and stochastic components.

**Methods And Evaluation Criteria:**

Yes.

**Other Comments Or Suggestions:**

1. Too many typos occurs in the submission, e.g. Conclusions Para.2.
2. Where is definition of $\mathcal{D}_{\theta}$? I highly recommend to clarify it in the main context.

**Other Strengths And Weaknesses:**

Weakness.
1. In my opinion, VAE is a more natural choice for stochastic encoding; however, the authors employ a deterministic encoder and subsequently inject noise into the latent variable. I find the intuition behind this approach questionable, as the explanation provided in the paper is not entirely convincing to me.
2. The novelty of the paper is questionable given the well-established connection between flow matching and diffusion modeling. The proposed approach bears a strong resemblance to CorrDiff, although the authors argue that CorrDiff employs a two-stage training process, where the regression encoder is trained first, followed by the diffusion of residual components. In contrast, this paper appears to adopt a joint training strategy, with the final loss formulated as the sum of both components.
3. Lack of detailed explanation of noise scaling, since it is one of the main contributions of the paper, as stated by the authors.

**Questions For Authors:**

See "Weaknesses".

**Relation To Broader Scientific Literature:**

The paper's contribution lies in applying the well-established techniques of flow matching and denoising diffusion modeling to enhance the resolution of small-scale details in natural images, particularly in weather science. While similar to CorrDiff, the proposed method introduces joint training of a regression encoder and residual diffusion, effectively mitigating potential overfitting issues that may arise in the two-stage training approach adopted in CorrDiff.

**Theoretical Claims:**

Yes, I checked all the mathematical derivations and proofs, including Appendix B and C for the derivation of denoising objective.

---

> ### Author Rebuttal · Authors · 2025-03-31
>
> We thank the reviewer for their constructive comments. Below are our responses to your remarks.
>
> *1. In my opinion, VAE is a more natural choice ...* (we shorten the questions due to the character limit)
>
> **Response.** Thank you for the insightful question. As clarified in our Remark in Sec 4.2, while VAEs are a natural choice for stochastic encoding, we intentionally opt for a maximum-likelihood-based approach with a deterministic encoder and post-hoc noise injection. This choice is motivated by both **simplicity** and **interpretability**.
>
> From a modeling standpoint, our approach yields a closed-form expression for the noise variance—simply the RMSE of the regression error—which enables straightforward tuning using validation data to control generalization. In contrast, VAEs require careful balancing of the KL term and reconstruction loss, which is known to be challenging and prone to issues such as posterior collapse and the prior hole.
>
> From a physics perspective, particularly in applications like weather downscaling, large-scale structures are largely deterministic. A deterministic encoder is therefore well-suited to capture these coherent features, while the stochasticity in small scales is effectively modeled via calibrated noise.
>
> While we agree that VAE-style encoders are a valid alternative, our design provides a principled and practical path that aligns well with physical intuition and avoids some known drawbacks of VAEs.
>
> ---
>
> *2. The novelty of the paper is questionable...*
>
> **Response.** We respectfully disagree. While our method shares surface similarities with CorrDiff, AFM introduces key innovations that address fundamental limitations of CorrDiff and generalize it as a corner case.
>
> *Joint Training vs. Two-Stage*: CorrDiff relies on a two-stage pipeline, first training a deterministic encoder, then fitting a diffusion model to the residuals. This separation causes overfitting in the encoder. In consequence, the residuals shrink and become uninformative, leaving the generative model. In contrast, AFM performs joint training of the encoder and the generative model, maintaining informative residuals throughout optimization.
>
> ** Uncertainty Modeling**: CorrDiff lacks an uncertainty-aware encoder. AFM introduces an adaptive noise injection mechanism that dynamically increases the encoder noise when the validation error rises, signaling overfitting. This prevents collapse and encourages the encoder to retain uncertainty.
>
> ** Channel-wise Adaptivity**: AFM performs per-variable noise scaling—crucial for scientific data where variables (e.g., temperature vs. radar reflectivity) exhibit different stochastic behaviors. CorrDiff applies uniform noise, which fails to capture such heterogeneity.
>
> These design choices lead to significantly improved calibration and spread-skill performance (see Tables 2–3, 6, 13; Figures 5–6, 8).
>
> ---
>
>
> *3. Lack of detailed explanation of noise scaling...*
>
> **Response.** Thank you for the remark. We have expanded our explanation of the noise scaling mechanism in Sec. 4.2, Appendix E, and Algorithm 1 of the revised manuscript.
>
> Briefly, we estimate $\sigma_z$ using a maximum-likelihood criterion linked to the encoder's RMSE on validation data (see Eq. 8). As training progresses, if the encoder begins to overfit—reflected by rising validation error—our method increases $\sigma_z$, injecting more uncertainty into the latent space. This prevents the encoder from collapsing into a deterministic solution and ensures that the flow-matching model remains calibrated and robust to out-of-distribution inputs.
>
> Unlike fixed or manually tuned noise levels, our approach adaptively adjusts $\sigma_z$ per variable (channel-wise), which is crucial for scientific data where different physical variables exhibit distinct noise scales and predictability (e.g., temperature vs. radar reflectivity, as shown in Fig. 5 and Fig. 8).
>
> In practice, to avoid abrupt changes, we update $\sigma_z$ using an exponential moving average (EMA) every $M=10{,}000$ training steps $\sigma_z \leftarrow (1{-}\beta)\, \sigma_z + \beta\, \sigma_z^{\mathrm{cur}}$
> The final $\sigma_z$ is fixed for inference. An ablation in Sec. 5.3 (Table 4) demonstrates the effectiveness of this adaptive noise scaling in improving ensemble calibration and spread-skill alignment.
>
> ---
>
> *4. Too many typos...*
>
> **Response.** Thank you for noticing the typos. We have corrected them in the revised manuscript.
>
> ---
>
>
> *5.Where is definition of  $\lambda $?..."*
>
> **Response.** Thank you for your comment. Currently  $\lambda $ is introduced in Sec. 4.3. It balances the deterministic regression and stochastic diffusion terms. In the limit  $\lambda \rightarrow \infty $, AFM reduces to a purely deterministic model. We have added this clarification to the main text in the revised manuscript.
>
> ---
>
> We hope our responses and revisions have adequately addressed your concerns.

---

### Official Review · Reviewer_xRHs · 2025-03-13

**Overall Recommendation:** 3

**Summary:**

The paper focuses on tackling small-scale physical science problems (e.g., weather super-resolution). It proposes the joint encoder and flow-matching training objective over the prior two-stage methodologies to improve the overfitting. Specifically, this work introduces Adaptive Flow Matching (AFM) with the help of adaptive noise scaling to introduce stochasticity in the process. At last, authors compare w.r.t. Various baselines on regional downscaling and multiscale Kolmogorov-Flow tasks.

**Claims And Evidence:**

In the abstract itself, it is claimed that AFM achieves SOTA on regional downscaling. However, this is not the case. According to Table 2 and Figure 2, improvements over CFM are arguable. However, on Kolmogorov-Flow, task performance seems to be much better. But overall, it feels marginally better.

**Essential References Not Discussed:**

N/A

**Experimental Designs Or Analyses:**

The reviewer verified all experimental designs and analyses. There are no visible issues.

**Methods And Evaluation Criteria:**

Methods: Yes, the proposed method makes sense, as spatially misaligned input/output could be challenging for the existing generative model formulations.

Evaluations: Due to a lack of reviewer expertise, it is hard to verify the evaluation procedure. However, it seems exhausting.

**Other Comments Or Suggestions:**

- Paper often misrepresents the Flow matching as a stochastic process (Abstract Line 25), which is misleading. Meanwhile, stochasticity comes from noise term in eq. (5).
- Typos in Figures 2,3 & 4 where SFM is written instead of AFM.
- “scientific” type in conclusion

**Other Strengths And Weaknesses:**

Weaknesses:
- The paper contains many typos and lacks clarity (see Suggestions).
- As flow matching maps the any to any distribution, there should be a baseline that just does that along with traditional CFM. Basically, direct $y$ to $x$ mapping. Similarly, other methods, such as “Diffusion Schrödinger bridge matching,” could also be added.
    - Because of these prior works (and missing baselines), I am unsure if the claim for Section 4 Line 149-152 is valid.

**Questions For Authors:**

- What is the final $\sigma_z$ value during the inference? Is it the one result at the end of training (Figure 5)?

**Relation To Broader Scientific Literature:**

The proposed approach to AFM might be of interest to the broader community as it attempts to perform joint training of the encoder and the flow model.

**Theoretical Claims:**

The review has verified Proposition 1  and the resulting sampling procedure. Both seem to be accurate.

---

> ### Author Rebuttal · Authors · 2025-03-31
>
> We thank the reviewer for their comments and for finding our paper interesting.
>
> *1. In the abstract itself, it is claimed that AFM achieves SOTA on regional downscaling. However, this is not the case. According to Table 2 and Figure 2, improvements over CFM are arguable. However, on Kolmogorov-Flow, task performance seems to be much better. But overall, it feels marginally better.*
>
> **Response.**
> We thank the reviewer for their remark. Our claim of “SOTA” is based on the consistent gains observed across datasets, channels and metrics, particularly in those with greater stochastic variability. In Table2, for instance, *radar reflectivity* sees a clear improvement when using AFM over CorrDiff, and AFM’s ensemble calibration (e.g., SSR) is superior across all channels. Similarly in Table 13 AFM outperforms all other models in most metrics and channels and has better calibration by a large narging.
> Also, in the Kolmogorov Flow experiments (Table 3), AFM clearly outperforms CFM and CorrDiff and the gap become wider as the misalignment increases (tau 3 -> 10). While the margins may appear moderate on more deterministic channels (e.g., temperature), these results overall indicate that AFM consistently delivers better calibration and uncertainty modeling, which is crucial in scientific tasks. We clarify in the revised abstract that our method excels especially under higher stochasticity and improves overall ensemble calibration.
>
> ---
>
> *2. As flow matching maps the any to any distribution, there should be a baseline that just does that along with traditional CFM. Basically, direct $\mathcal{X} \to \mathcal{Y}$ mapping. Similarly, other methods, such as Diffusion Schrödinger bridge matching,' could also be added. Because of these prior works (and missing baselines), I am unsure if the claim for Section 4 Line 149-152 is valid.*
>
> **Response.** Thank you for the comment. In our setting, direct $\mathcal{X} \to \mathcal{Y}$ flow matching is not applicable due to both spatial and channel misalignments—$\mathcal{X}$ and $\mathcal{Y}$ often have different modalities (e.g., $\mathcal{X}$ includes forecast features, while $\mathcal{Y}$ includes precipitation and radar reflectivity). As such, intermediate alignment via an encoder is necessary. The closest viable baseline is Conditional Flow Matching (CFM), which we include. Methods like diffusion Schrödinger bridges (e.g., I2SB) also assume aligned domains and are similarly inapplicable here.
>
> ---
>
> *3. Paper often misrepresents the Flow matching as a stochastic process (Abstract Line 25), which is misleading. Meanwhile, stochasticity comes from noise term in eq. (5).*
>
> **Response.** Thank you for pointing this out. You are correct—flow matching is inherently deterministic (ODE-based), and the stochasticity in our method arises from the noise injected into the encoder output. We have clarified this distinction in the abstract of the revised manuscript.
>
> ---
>
> *4.Typos in Figures 2,3 & 4 where SFM is written instead of AFM. “scientific” type in conclusion*
>
> **Response.**
> Thank you for noticing the typos, that are now fixed in the revised manuscript.
>
> ---
>
> *5. What is the final $\sigma_z$ value during the inference? Is it the one result at the end of training (Figure 5)?*
>
> **Response.**  Yes, that is correct. During inference, we use the final converged value of \(\sigma_z\) obtained at the end of training, as shown in Figure~5.
>
> We hope our responses and revisions have adequately addressed the remaining of your concerns.

---

> > ### Comment · Reviewer_xRHs · 2025-04-01
> >
> > I thank the authors for providing the clarifications. Here is my updated response:
> >
> > 1. (Q1) I acknowledge that the proposed method indeed shows improvements over the baseline (as also mentioned in the original review). However, performance again is not significant despite limited novelty (also ack. by other reviewers). Additionally, due to the reviewer's lack of data domain expertise, I will assume that this is good enough.
> >
> > 2. (Q2) Thanks for the clarification. What if we add the extra random channel to match the dimensionality?
> >
> > To summarize, I will maintain my current score with 3/5 confidence.

---

> > > ### Author Response · Authors · 2025-04-03
> > >
> > > We thank the reviewer for the follow-up. In our CWA experiments, the input has 20 channels while the target has only 4, so dimension matching via random auxiliary channels is not feasible. More importantly, matching dimensionality alone is not sufficient—the key issue is that the content of the input and target channels must be compatible. Since flow matching relies on linear interpolation in data space, mixing unrelated modalities (e.g., temperature in the input and radar reflectivity in the target) leads to semantically meaningless interpolants and poor learning behavior.
> > >
> > > Thus, an encoder is essential to map the input to a representation aligned with the target distribution, making their content mixable and interpolation meaningful. This was the motivation behind our initial point: methods like direct flow matching or Schrödinger bridges assume spatial and semantic alignment between source and target, which does not hold in our setting.

---

### Official Review · Reviewer_eJPb · 2025-03-13

**Overall Recommendation:** 4

**Summary:**

The paper addresses image 'super-resolution' in the context of atmospheric physics. The super-resolution aims at generating small stochastic scales into the input data while preserving and aligning large scale physics. The authors propose to adapt and apply flow matching for this purpose. To this end, the authors show that it is possible to jointly train a time dependent decoder and the flow. An adaptive noise scaling factor allows to adjust the noise level of the diffusion process to the input data Experimental results are illustrated on two meteorological datasets. Quantitative results and comparisons with related work are reported.

"## update after rebuttal
The rebuttal provided by the authors has addressed my few concerns. The paper has probably a limited audience amongst the ML/CV community, but nevertheless illustrates a very relevant practical use case (and associated challenges) beyond the known benchmarks.

**Claims And Evidence:**

The authors claim that small physics scales can be accounted for with a diffusion denoising model while the large scale discrepancy between output/input can be tackled with an determinist encoder. Experimental setting and comparison with standard approaches demonstrate promising results.

**Essential References Not Discussed:**

no missing reference, to my knowledge.

**Ethical Review Concerns:**

no ethical concern

**Experimental Designs Or Analyses:**

Experimental results are limited but convincing. One real data and one synthetic data are used for training and evaluation. Details of the datasets are reported in appendix. Thorough and detailed visual and quantitative results are reported. It is interesting in particular to observe the discrepancy of quantitative results for stochastic channels (ie radar and wind) vs deterministic (ie temperature) channels.

**Methods And Evaluation Criteria:**

The motivation of the approach is clearly exposed and the mathematical derivation of the problem clearly presented. The steps of the  approach and reasoning are well justified and discussed.
Performance is evaluated in terms of RMSE, Continuous Ranked Probability Score (CRPS), Mean Absolute Error (MAE), and Spread Skill Ratio (SSR).Ablation studies are performed. Thorough comparisons are reported.

**Other Comments Or Suggestions:**

- In figure 2, 4: there is a confusion between the acronyms: SFM -> AFM
- The authors could be more specific when describing the "mis-alignment" between the input / output. How this mis-alignment could be characterized: is it global or local, subpixel or few pixels scale?
- I understand that the input low resolution data is first rescaled by bilinear interpolation to the size of the output. Is it correct?

**Other Strengths And Weaknesses:**

A well presented paper, with a clear 'flow' , a well justified approach, and limited but convincing results.

**Questions For Authors:**

Could you envisage different fields of application? Could this approach be useful for other use cases?

**Relation To Broader Scientific Literature:**

This paper is at the frontier between applied atmospheric physics and machine learning. It introduced most relevant material, without being exhaustive, as required.

**Theoretical Claims:**

The mathematical derivations (main paper and supplementary material) are correct tmk

---

> ### Author Rebuttal · Authors · 2025-03-31
>
> We thank the reviewer for their positive and constructive feedback, and for recognizing our clear motivation and rigorous methodology.
>
> *1. In figure 2, 4: there is a confusion between the acronyms: SFM -> AFM*
>
> **Response.**
>  Thank you for noticing the typo. In the revised manuscript, we have used AFM consistently.
>
> *2. The authors could be more specific when describing the 'misalignment' between the input / output. How this mis-alignment could be characterized: is it global or local, subpixel or few pixels scale?*
>
> **Response.**
> Thank you for the suggestion. We have clarified the notion of misalignment in the revised manuscript. It arises in two ways: (1) channel-level mismatch, where input and output contain different modalities (e.g., forecasts vs. radar reflectivity); and (2) spatial misalignment due to differences in the underlying PDEs between coarse and fine models. Spatially, the misalignment can be both global and local. For instance, in Fig. 7, storm centers in wind fields are displaced by several grid points (large-scale shift), while in Fig. 8, smaller PDE discrepancies ($\tau{=}3$) lead to subpixel-to-few-pixel level local misalignment.  In the revised manuscript we have added a section G.3 discussing this in further detail and also adapted Section 5 to more clearly define misalignment.
>
> *3. I understand that the input low resolution data is first rescaled by bilinear interpolation to the size of the output. Is it correct?*
>
> **Response.**
> Thank you for the question. Indeed, we apply bilinear upsampling to align dimensions, then let the encoder refine and align further. This is explained in Section 5.1 of the manuscript (and in more detail in G.1 of the Appendix). In the revised version we added further clarifications regarding this in the main text (Sec 5.1).
>
> *3. Could you envisage different fields of application? Could this approach be useful for other use cases?*
>
> **Response.**
> Certainly. Our AFM framework applies wherever partial or misaligned data requires super-resolution or channel synthesis, e.g., MRI subsampling in medical imaging. We highlight broader applicability in the Conclusion of the revised manuscript.
>
> We hope our responses and revisions have adequately addressed your questions.

---

### Decision · Program_Chairs · 2025-05-01

**Decision:**

Accept (poster)

**Comment:**

This paper presents an adaptive flow matching approach for defining a super-resolution method adapted to small-scale physics. An application is proposed on a real world weather dataset in addition to synthetic Kolmogorov flow datasets.

During the evaluation strong points have been raised: the method is well presented and discussed, improvement convincing while limited, the paper illustrates a relevant practical use case.
The following negative points have also been noticed: limited experimental evaluation, other baselines could have been considered, differences unclear with other methods notable CorrDiff.

During rebuttal, authors have answered to reviewers' remarks, precising better the scope of the contribution and differences with other methods (notably CorrDiff).
After the rebuttal and discussion, reviewer zyDu improves his score to weak accept, reviewers xRHs and X7XZ maintained their scores to weak accept and reviewer eJPb kept his score as accept.

While a part of the methodology and some experiments were evaluated as a bit limited, the contribution, there was a consensus for saying that the results are convincing, the method is sound and well justified and the contribution can be interesting for a broad audience. The positive points outweigh the negative ones.
I propose then acceptance.